



# Modelling the European wind-blown dust emissions and their impact on PM concentrations

Marina Liaskoni[1], Peter Huszar[1], Lukáš Bartík[1], Alvaro Patricio Prieto Perez[1], Jan Karlický[1], and Ondřej Vlček[2]

[1]Department of Atmospheric Physics, Faculty of Mathematics and Physics, Charles University, Prague, V Holešovičkách 2, 18000, Prague 8, Czech Republic
[2]Czech Hydrometeorological Institue, Na Šabatce 2050/17, 143 00 Prague 12, Czech Republic

**Correspondence:** Marina Liaskoni (marina-despoina.liaskoni@matfyz.cuni.cz)

**Abstract.** Wind-blown dust (WBD) emitted by the Earth's surface due to sandblasting can potentially have important effects on both climate and human health via interaction with solar and thermal radiation and reducing air-quality. Apart from the main dust "centers" around the world like deserts, dust can be emitted from partly vegetated middle and high latitude areas like Europe if certain conditions are suitable (strong winds, bare soil, reduced soil moisture, etc.). Using a wind-blow dust
model (WBDUST) along with a chemical transport model (CAMx) coupled to a regional climate model (WRF), this study as one of the first ones provides a model based estimate of such emissions over Europe as well as the long-term impact of WBD emissions on the total PM concentrations for the 2007-2016 period.

We estimated WBD emissions to about 0.5 and 1.5 $\mathrm{Mg\,km^{-2}\,yr^{-1}}$ in fine and coarse mode in average. Maximum emissions occur over Germany where the average seasonal fine and coarse mode emission flux can reach 0.2 and 0.5 $\mathrm{g\,km^{-2}\,s^{-1}}$, respec-
tively. Large variability is seen in the daily averaged emissions with values up to 2 $\mathrm{g\,km^{-2}\,s^{-1}}$ for the coarse mode aerosol on selected days.

The WBD emissions increased the modelled winter PM2.5 and PM10 concentrations by up to 10 and 20 $\mathrm{\mu gm^{-3}}$, respectively, especially over Germany, where the highest emissions occur. The impact on other seasons is lower. Much higher impacts are modelled however during selected days when occasionally the urban PM2.5 and PM10 concentrations are increased by more
than 50 and 100 $\mathrm{\mu gm^{-3}}$. The comparison with measurements revealed that if WBD is considered, the summer biases are reduced however the winter PM is even more overestimated (so the bias increased). We identified strong overestimation of the modelled wind-speed (the maximum daily wind is almost 2 times higher in WRF than the measured ones) suggesting that WBD emissions are also overestimated hence the enhanced winter PM biases.

Moreover, we investigated the secondary impacts of the crustal composition of fine WBD particles on secondary inorganic
aerosol (SIA): sulphates (PSO4), nitrates (PNO3) and ammonium (PNH4). Due to perturbing the water pH value and thus the uptake of their gaseous precursors as well as due to increased aerosol surface serving as oxidation site, we modelled increased seasonal PSO4 and PNO3 concentrations by up to 0.1 $\mathrm{\mu gm^{-3}}$ and decreases for PNH4 (by up to -0.05 $\mathrm{\mu gm^{-3}}$), especially during winter. As the average daily impact, these numbers can however reach much larger values up to 1-2 $\mathrm{\mu gm^{-3}}$ for sulphates and nitrates while the decrease of ammonium due to WBD can reach -1 $\mathrm{\mu gm^{-3}}$ on selected days. The sensitivity





test on the choice of the inorganic equilibrium model (ISORROPIA vs. EQSAM) showed that if EQSAM is used, the impact on SIA is slightly stronger (by a few 10%) due to larger number of cations considered for water pH in EQSAM.

Our results have to be considered as a first estimate of the long-term WBD emissions and the related effects on PM over Europe. More sensitivity studies involving the impact of the WBD model choice and the input data used to describe the land-surface need to be carried out in future to better constrain these emissions.

## 1 Introduction

Wind-blown dust (WBD) emitted by the Earth's surface can have a significant effect on both climate and human health by reducing air quality. It affects the climate directly and indirectly by scattering solar radiation, modifying the cloud properties and precipitation as it can also serve as cloud-condensation nuclei (Ryder at al., 2013; Song et al., 2022). Additionally, exposure to high levels of dust particles can have severe effects on human health concerning the respiratory and the cardiac system (Giannadaki et al., 2014; Keet et al., 2018).

One of the major emitters of WBD of concern in Europe (but also globally) is the Sahara desert which contributes to 50% of dust emissions globally. Sahara dust is a major contributor to European atmospheric pollution as well, and its levels are critically high in southern Europe, while light dust episodes are often detected above Central and Western Europe (Wang et al., 2020). Other natural sources can be wildfires, which due to intense turbulence can generate dust emissions (Wagner et al., 2018). WBD can be emitted also by non-vegetated areas containing fine and loose sediments when strong winds occur. Human activities contribute significantly in the increasing dust generation too. Destruction of soil crust and vegetation removal in semi-arid regions, changing cultivation patterns and new transport pathways are some of the most affecting anthropogenic activities (Birmili et al., 2008).

During climate change, dust emissions are anticipated to increase in the future (Zittis et al., 2022). Modified climate conditions (with the associated weather patterns) and changes in land use are the main affecting factors. If dry periods between the precipitation events get prolonged, then the soil of the surface is going to be prone to strong winds, ending up in an increase in dust emissions. Gudmundsson et al. (2016) assessed how the anthropogenic contribution to the emissions has affected the probability of droughts in Europe. Their results stress that the drought risk for southern Europe has already increased, although the results for Central Europe are inconclusive. Stagge et al. (2017) used two precipitation indices and showed significant increases in drought likelihood for southern Europe, and decreases in the total area of the north, resulting in values that are dependent on the geographical domain, and can shift the spatially-averaged values for whole Europe. On the other hand, many studies have shown that fine particles can be transported over long distances through the atmosphere and elevate PM levels in different areas of the continent, far from the area of source (Ansmann et al., 2003; Francis et al., 2022; Zwaaftink et al., 2022). Hence, mineral dust emissions must be examined in connection to both the main dust centres (like the Sahara) but also those emitted over non-arid areas like Europe where normally the temporal distribution of precipitation and denser vegetation prevents the necessary soil drying for such emissions. Indeed, as said above, within a changing climate, such conditions can be more frequent.



Over Europe, very few studies accounted for the local (i.e. not that advected from other continents) dust emissions. Recently, Meinander et al. (2022) identified potential dust sources over Europe (among other areas). Korcz et al. (2008) gave a detailed

model based estimate for the spatial and temporal variation of such emission using a mesoscale weather model (MM5) as the meteorological driver. They however did not compute their contribution to the total PM concentrations. Vautard et al. (2005) calculated the emission from natural erosion and re-suspension over Europe and found significant model (CHIMERE) improvement when these emissions are accounted for PM they however considered only two seasons in a selected year without giving account for long-term effects. Similarly, Bessagnet et al. (2008) considered a strong European dust event originating

in Ukraine, but this cannot be considered representative for long-term. Recently, Kakavas and Pandis (2021) looked at urban dust over Europe and calculated its impact on PM levels. Moreover, they also accounted for the impact on the formation of secondary aerosols. They showed that the urban dust source can be significant and can potentially reduce model biases. However, they were not interested in other dust sources, e.g. those. originating from soils from rural/natural areas and they looked only at one month not providing a long term estimate.

Motivated by this, here we present a novel study to quantify the long-term dust emissions for present-day conditions over central Europe using a regional climate model coupled with a chemical transport model along with a WBD model for dust fluxes. For the correct modelling of potential future evolution of WBD it is crucial to first evaluate the models ability to resolve their present-day magnitude and the associated impact on the total PM concentrations. Our study focuses on the long-term impact during a 10y period which allows us to obtain a representative pattern of the temporal and spatial distribution of the

WBD emissions and their overall impact on PM levels. Moreover, this study will also look at the secondary impact of WBD particles on secondary aerosol components focusing on the inorganic aerosol. Indeed, there is an indication that the composition of dust particles can have an indirect impact on nitrates, sulphates and ammonia (Fairlie et al., 2010; Karydis et al., 2011; Wang et al., 2012; Malaguti et al., 2015; Kakavas and Pandis, 2021; Wang et al., 2022) either by acting as a surface for heterogeneous reactions (e.g. Fu et al. (2016); Wang et al. (2022)) or by their ion composition and modulating aqueous reactions that form

nitrates and sulphates (Kakavas and Pandis, 2021) representing an indirect pathway of impacting the overall PM levels. In this study, the main interest will be the quantification of WBD contribution to urban PM levels as urban areas already experience adverse air-pollution episodes and it is of interest to calculate how natural emissions like WBD can potentially contribute to urban PM concentrations.

## 2   Methods and data

To achieve the goals of the study, we applied the chemical transport model CAMx driven offline by regional climate model WRF. The emissions of wind-blown dust were calculated by the WBDUST emissions model. All these models are described in detail below.





## 2.1 Dust model

The dust emission scheme WBDUST used here is based on Klingmüller et al. (2018) study, who updated a new dust emission
scheme based on Astitha et al. (2012)'s approach. This scheme combines meteorological parameters with descriptions of land
cover type, clay fraction of the soil, the vegetation cover, the topography factor and the chemical composition. From the
landcover data, "barren or sparsely vegetated" grid fractions are identified as land capable of dust emissions. The clay fraction
is used to calculate the sandblasting efficiency which increases exponentially with a clay fraction up to 20%, beyond that it is
considered constant. Another important parameter influencing the dust emissions is the amount of vegetation. Quantitatively it
is expressed as the total area of the leaves relative to the surface area called Leaf Area Index(LAI). In the WBDUST model, no
emissions are considered for LAI > 0.35 while full emissions occur at zero LAI with linear dependence between. In the dust
module, LAI is converted to the vegetation factor (fveg) defined as:

$$fveg = 1 - \frac{min(LAI, 0.35)}{0.35} \tag{1}$$

Consequently, the vegetation factor takes values between zero and one, where zero value corresponds to full emissions (no
vegetation) and one means no emissions (i.e. full vegetation). To avoid the situation where the average LAI over a gridcell
is higher than 0.35 leading to zero dust emissions, although the gridcell may contain fractions with lower LAI that would
otherwise emit some dust, we first converted LAI data into fveg data retaining the same resolution. Only after this step we
redistributed it into the model gridcell. With this approach, we accounted for the dust emittable surface fractions with limited
vegetation. The maps on Fig. 1 represent the WBDUST input data, namely the clay fraction and the LAI converted vegetation
factor for January and July, taken from the middle of the decade of interest (year 2010).

The emission flux for dust in size mode "i" in WBDUST is calculated by the following equation:

$$j_{emis,i} = \frac{c\rho_{air}}{g}(u_* + u_{*t})^2(u_* - u_{*t})10^{-4}\alpha f_{landcover} f_{veg} M_i N S_{topo}, \tag{2}$$

where c is an empirical constant (here = 1.5), $u_*$ is the surface friction velocity, $u_{*t}$ is the threshold surface friction velocity,
$f_{landcover}$ is the barren land fraction, $f_{veg}$ is the vegetation factor, $\alpha$ is the sandblasting efficiency, $\rho_{air}$ is the air density, $g$
is the gravitational acceleration, $M_i$ is the mass fraction emitted into the mode "i", $N$ is the normalization factor and finally,
$S_{topo}$ is the topography factor parameter, which enhances the representation of emissions which are generated in valleys and
basins. The equation for the threshold surface friction velocity can be found analytically in Klingmüller et al. (2018).

WBDUST is based on Fortran and is provided as a preprocessing tool along with the CAMx code (https://www.camx.com/
download/support-software/, last visited 30 Nov 2022). It is driven by WRF meteorological data (see below) while the required
parameters are described in Section 2.4.



## 2.2 Driving meteorological model

To drive the dust model with meteorological data as well as to drive the chemical transport model used, the WRF (Weather Research and Forecasting) model version 4 (Skamarock et al., 2019) was used. In WRF, the radiation processes are parameterized by RRTMG scheme (Iacono et al., 2008), microphysical processes and convection were treated by Purdue Lin scheme (Chen and Sun, 2002) and the Grell-3D scheme (Grell, 1993), respectively. The description of surface layer processes followed the Eta model (Janjic, 1994). The land surface exchange is parameterized by the Noah (Chen and Dudhia, 2001) and, finally, the boundary-layer is resolved by the BouLac planetary boundary layer scheme (Bougeault and Lacarrère, 1989). Static land-use data for WRF is derived from CORINE Land Cover data, version CLC 2012 (CORINE, 2012). For urban grid-boxes, the single-layer urban canopy model (SLUCM;(Kusaka et al., 2001)) is used with the same urban parameters as in Karlický et al. (2018). The choice of physical parameterizations is based on results from Karlický et al. (2020) who performed a series of sensitivity experiments to achieve the best possible model-observation agreement. To drive the regional climate in WRF, the ERA-interim reanalysis (Simmons et al., 2010) was used.

## 2.3 Chemical transport model

To account for the transport of the emitted dust and its interaction with the aerosol physical and chemical processes we used the chemical transport model CAMx version 7.10 (Comprehensive Air-quality model with Extensions; Ramboll et al. (2020)). CAMx is an Eulerian chemical transport model that simultaneously treats photochemistry and aerosol processes. As gas-phase chemistry and secondary aerosol formation are closely linked and, moreover, in our study we are interested in the impact of dust on secondary inorganic aerosol, we considered the "full" gas-phase chemistry in CAMx using the CB6r5 mechanism (Carbon Bond revision 6) described in Yarwood et al. (2010) and Emery et al. (2015).

For aerosol, a static two-mode (fine/coarse) approach called CF2E is adopted. Secondary inorganic aerosol is partitioned between gas and aerosol phases using either the ISORROPIA thermodynamic equilibrium model v1.7 (Nenes et al., 1998, 1999) or the EQSAM (EQuilibrium Simplified Aerosol Model V4) model (Metzger et al., 2016). ISORROPIA considers sulphate (PSO4), nitrate (PNO3), chloride (NCL), ammonium (PNH4), and sodium (NA), with an update for calcium nitrate on dust particles, which is important for our study. Aqueous nitrate and sulphate formation in cloud water is computed using the RADM-AQ aqueous chemistry algorithm (Chang et al., 1987) with updated sulphur dioxide ($SO_2$) oxidation reaction rates and metal-catalysed oxidation mechanism. A semi-volatile equilibrium scheme called SOAP (Strader et al., 1999) is used to form secondary organic aerosol from condensable vapours.

Apart from the secondary (in)organic aerosol, primary elemental (PEC) and organic carbon (POA), CAMx further considers general primary aerosol categories for fine crystal (dust; FCRS) and other fine primary aerosol (FPRM) and also for their coarse counterparts (CCRS and CPRM). The two mode CF2E approach optionally includes eight explicit fine-mode elemental species iron (Fe), manganese (Mn), calcium (Ca), Mg, K, aluminium (Al), silicon (Si) and titan (Ti) which can be either modelled or background values are used for chemical calculations. Calcium is an exception, which is scaled from FCRS and FPRM.





The species FPRM, FCRS, CPRM, CCRS including the 8 elements do not chemically decay. However, light scattering by them along with other PM components affecting photochemistry is considered. Furthermore, the fine-mode species concentra-

tions influence PM and heterogeneous gas chemistry. In RADM-AQ, the oxidation of $SO_2$ to sulphate is catalytically enhanced by Fe and Mn while Mg, Ca and K affect cloud pH hence the solubility of $SO_2$. Further Mg, Ca and K influence inorganic aerosol partitioning in EQSAM and Ca reacts with $HNO_3$ soil dust particles to form calcium nitrate ($CaNO_3$) in ISORROPIA. Fine aerosol species FPRM and FCRS along with the 8 elements represent surface area for heterogeneous reactions of $SO_2$ and $N_2O_5$. Uptake of $SO_2$ and $HNO_3$ by dust particles is also considered using a humidity-dependent uptake coefficient (Zheng et

al., 2015).

CAMx was driven using WRF output translated to CAMx meteorological inputs using the wrfcamx preprocessor that is supplied along with the CAMx code http://www.camx.com/download/support-software.aspx (last access: 25 Nov 2022). The coefficients of vertical eddy-diffusion (Kv) are diagnosed in wrfcamx based on the similarity method adopted from the CMAQ model (Byun and Ching, 1999). The choice of the method for the calculation of Kv is crucial as it greatly determines the

species vertical transport, especially over urban environments (Huszar et al., 2020). They further showed that the CMAQ method represents a mid-range of the Kv intensities diagnosed from WRF output.

## 2.4    Experiments and data

A series of model simulations using CAMx coupled offline to WRF were carried out over a "larger" central European domain of size 189 × 165 (from France to Ukraine, northern Italy to Denmark) at 9 km × 9 km horizontal resolution centered over

Prague (Czechia) (50.075N, 14.44E, Lambert conic conformal projection). WRF has 40 layers in vertical reaching 50hPa with the lowermost layer about 30 m thick. CAMx uses 18 layers with the top one at about 10 km. As long term impact of WBD emissions is analysed here, we covered a 10yr simulation period from 2007/01/01 to 2016/12/31.

As already said, WRF was driven with the ERA-Interim reanalysis while for CAMx chemical initial and boundary conditions we choose the CAM-Chem global model data (Buchholz et al., 2019; Emmons et al., 2020).

As anthropogenic emissions, the TNO-MACC-III data (an update of the MACC-II version; Kuenen et al. (2014)) were used from 2011 for the whole period. This high resolution (1/8° longitude 1/16° latitude, roughly 6 km x 6 km) European emission database provides annual emission estimates for NOx, $SO_2$, non-methane volatile organic compounds (NMVOC), methane ($CH_4$), ammonia ($NH_3$), carbon monoxide (CO) and PM10 and PM2.5 in 11 activity sectors. The annual emission totals were redistributed to model grid-cells using the FUME (Flexible Universal Processor for Modeling Emissions) emission model

(Benešová et al. (2018), http://fume-ep.org/, last access: 1 September 2022). FUME also took care of chemical speciation and time disaggregation of input, sector based emissions while the speciation and time disaggregation factors are based on Passant (2002) and van der Gon et al. (2011). The output of the FUME is CAMx-ready hourly emission data for the speciated model species. Biogenic emissions for CAMx are calculated offline with the MEGANv2.1 (Model of Emissions of Gases and Aerosols from Nature) model (Guenther et al., 2012) based on WRF meteorology and vegetation characteristics following

Sindelarova et al. (2014).



For the WBDUST module, the inputs were the following. The land cover was described using the high resolution (100 m) CORINE CLC 2012 land cover data (https://land.copernicus.eu/pan-european/corine-land-cover; CORINE (2012)) in combination with the United States Geological Survey (USGS) database for gridcells with no information from CORINE. This landuse was used also for CAMx dry-deposition scheme. The clay fraction data comes from the Global Soil Dataset for use in

Earth System Models (GSDE; Shangguan et al. (2014)). The GSDE provides the clay fraction of the topmost 4.5 cm soil layer, which is most relevant for the sandblasting efficiency. Leaf-area-index data are taken from MODIS post-processed data provided by Yuan et al. (2011) at 30" resolution (around 500 m over our domain) with 8 day update interval. Year 2010 LAI was used for the whole period. As topography information to calculate the topography factor, the Global Multi-resolution Terrain Elevation Data 2010 (GMTED, 2010) is being used, with a spatial resolution of $0.1°$.

One of the important goals of the study is to examine the potential impact of WBD elemental composition (Na+, K+, Fe+, Mn+, Ca++ and Mg++) onto the formation of secondary inorganic aerosol. Therefore, we must also consider the chemical soil composition of the emitted dust. We estimated it based on fractions that were calculated by Karydis et al. (2011).

The emissions of wind-blow dust (with the model described above) were calculated for fine and coarse crustal material based on WRF output meteorology: surface temperature; soil moisture, snow water equivalent, wind, temperature, pressure

and geopotential height of the two lowermost levels. WBD emissions were calculated thus on an hourly basis (in accordance with output frequency). The calculation was done for six elements (Ca, Fe, Mg, Mn, K and Na) while the mass fraction of fine dust that does not belong to any of the listed elements is emitted as general fine crustal material (FCRS). Coarse crustal material is also emitted as one general species CCRS.

In order to account for the sensitivity on the method for gas partitioning into the aerosol phase as well as due to the fact, that

the CAMx crustal elements interact with aerosol chemistry differently, we conducted CAMx experiments for both the ISORROPIA and EQSAM equilibrium models. With each of these, a pair of experiments was conducted: i) one without considering WBD (only anthropogenic aerosol source and anthropogenic- and MEGAN-based gas-phase emissions) and one ii) with WBD considered. The experiments are named accordingly ISORROPIA_noWBD, ISORROPIA_WBD, EQSAM_noWBD and EQSAM_WBD.

In our analysis, we will examine the impact of WBD on PM2.5 and PM10 concentrations evaluted based on the ISORROPIA experiment pair. The EQSAM pair of simulations will be used to analyse the sensitivity of the impact secondary aerosol chemistry. It is clear that if dust particles influence the heterogeneous aerosol chemistry, the total contribution of WBD will be not simply the sum of concentrations of FCRS and the listed elements, but instead, we have to account for the effect dust has on secondary aerosol. Therefore the impact will be calculated as follows:

$$\Delta PM2.5 = PM2.5_{WBD} - PM2.5_{noWBD}, \quad (3)$$

while $PM2.5_{WBD}$ is calculated as

$$PM2.5_{WBD} = PEC + POA + FPRM + PSO4 + PNO3 + PNH4 + SOA + FCRS + Ca + Fe + Mg + Mn + K + Na \quad (4)$$





$PM2.5_{noWBD}$ is calculated as

$$PM2.5_{noWBD} = PEC + POA + FPRM + PSO4 + PNO3 + PNH4 + SOA + FCRS, \tag{5}$$

while FCRS here stands for fine crustal material entering the domain trough boundaries (it is not directly emitted within anthropogenic sources). For the impact on PM10, we added CPRM and CCRS to these sums to account for the anthropogenic and dust coarse mode aerosol, i.e.:

$$\Delta PM10 = (CCRS_{WBD} + PM2.5_{WBD}) - (CCRS_{noWBD} + PM2.5_{noWBD}). \tag{6}$$

Regarding $CCRS_{noWBD}$, as CCRS is not emitted in the noWBD simulations, this accounts for the crustal material entering
the domain via the boundaries similar to FCRS above.

## 3 Results

### 3.1 Modelled WBD emissions

In this section, the dust emission fluxes calculated using the WBDUST emission module are analysed. The validation of the underlying meteorological conditions driving the emission model as well as the resulting PM concentrations are validated in
the next section.

In Fig. 2, the two maps represent the seasonal average emissions for winter, the season with highest emissions calculated. Winter averaged FCRS dust emissions have values that can reach up to 25 $\mathrm{gs^{-1}gridbox^{-1}}$ while CCRS dust emissions can reach values that exceed 40 $\mathrm{gs^{-1}gridbox^{-1}}$. Increased emissions are noticed above western Germany where many farmlands and agricultural areas are located. High emissions are also often concentrated around urban areas. Although the urban land–use
category is not considered as bare soil, at the used resolution, many of the urban grid boxes are only partly covered by urban land cover (only a very few grid cells are covered by urban land cover by more than 50%) and the rest is usually a crop land which is potentially capable of dust emissions. As in the used LAI input (MODIS), it is often the cities, which have the sufficiently low LAI values (less than 0.35), it is there and over surrounding areas, where the conditions for WBDUST emissions are met (low LAI and bare soil).
The seasonal variability was also assessed by calculating the average annual cycle of the monthly mean domain averaged emissions. Fig. 3 confirms that higher emissions occur in winter season for both FCRS and CCRS, while the main emitting period begins in October and ends in April, proving that the presence of winds along with low LAI are the governing factors for dust emissions.

The temporal variability of these emissions on a daily and hourly basis is shown in terms of the daily average values and the
average diurnal cycle, respectively. Fig. 4 represents the time-series of the domain averaged daily averages. A high variability of daily emissions is seen and FCRS emissions can exceed 25 $\mathrm{gs^{-1}gridbox^{-1}}$ while CCRS emissions can reach values higher than 100 $\mathrm{gs^{-1}gridbox^{-1}}$ on specific days.





Fig. 5 shows the average diurnal cycle of the average hourly emission fluxes for different seasons. Emissions peak during mid-day which is associated with stronger winds and usually lower stability enabling to lift the sandblasted soil to produce emissions. The daily amplitudes are about 0.5-1 $\mathrm{gs}^{-1}\mathrm{gridbox}^{-1}$ and 1-2 $\mathrm{gs}^{-1}\mathrm{gridbox}^{-1}$ for FCRS and CCRS, respectively.

## 3.2 Validation

### 3.2.1 Meteorological fields

As the modelled wbdust emissions depend on meteorological conditions and the state of the soil, it is important to evaluate how well the driving model (WRF) represents the meteorological conditions that affect emissions fluxes the most. In this section we compare the modelled temperature and wind speed with available measurements from the area of Czech republic, while the soil moisture will be compared with satellite data. Although it represents a small fraction of the entire domain, we expect that the model biases are representative for larger areas. Measured temperature and wind data are from 10 Automated Imission Monitoring (AIM; www.chmi.cz, last visited 30 NOV 2022) network air-quality monitoring stations that, besides air quality data, provide also the basic meteorological variables.

Starting with the temperature, Fig. 6 represents the seasonal 2007-2016 averaged diurnal cycles. It is clear that the daily maximum temperatures are underestimated by the model during summer (JJA) while better match is achieved in other seasons. The autumn (SON) data show some positive model bias too. Regarding daily minima, the model tends to overestimate it for summer and autumn while a clear underestimate occurs in winter (DJF). The above mentioned biases are always less than 2°C and usually less than 1°C.

As from dust emission perspective, the maximum wind speeds are more relevant than the average ones, we also compared the modelled monthly mean of the maximum daily wind speeds averaged over 2007-2016. Results are depicted in Fig. 7. It is clear that the model reasonably captures the annual cycle of wind with minima during late summer early autumn and maximum wind speeds during winter. However, a strong positive model bias is evident reaching 2-4 $\mathrm{ms}^{-1}$, except the Praha-Ruzyne station and Brno-Turany during summer.

Finally, the state of the soil in terms of moisture content is another key driver of emissions with low soil moisture promoting sand blasting and thus dust emissions. For this quantity, we used the ESA CCI SM v07.1 satellite based dataset (Dorigo et al., 2017; Gruber et al., 2019) and plotted the spatial distribution (for the area of Czechia) of the 2007-2016 seasonal means volumetric soil moisture, in Fig. 8. The satellite data shows a strong annual variation with minimum values during summer (0-0.2 $\mathrm{m}^3\mathrm{m}^{-3}$) while much higher values during summer (0.4-0.6 $\mathrm{m}^3\mathrm{m}^{-3}$). This annual cycle is seen also in the modelled data but is much weaker with summer soil moisture data slightly lower than the winter ones. It is also clear that the model overestimates the observed data, especially during summer while the winter overestimation is small (with model values around 0.4-0.5 $\mathrm{m}^3\mathrm{m}^{-3}$) with even some underestimation limited to small regions.





### 3.2.2 PM concentrations

In this section, our results will be validated by comparing the modelled PM2.5 and PM10 concentrations calculated by CAMx
(by the ISORROPIA experiment pair) with observations. The observations were retrieved from the European air quality
database - Airbase (https://discomap.eea.europa.eu/map/fme/AirQualityExport.htm, last visited 30 NOV 2022; EEA, 2021)
available "(sub)urban-background" stations from selected European cities (i.e. Vienna, Prague, Berlin, Munich, Budapest and
Warsaw). These observations were plotted along with WBD and noWBD CAMx concentrations daily averaged for the 6 European cities for 2007-2016.

Fig. 9 and 10 depict daily time series for modelled and measured PM2.5 concentrations for selected European cities. In
general, the time evolution of observed values is well captured by the model simulations. It is also seen that during summer
months, concentrations are usually underestimated. For winter, when the highest measured peaks occur (often exceeding 100
$\mu gm^{-3}$), the model often fails to correctly capture the strength of the peak or its timing. It is also clear (and expected ) that the
WBD simulation generates highest peaks which are closer to the observed peaks, or even exceeds those suggesting a positive
model bias during winter. For PM10 (Fig. 11 and 12) the situation is similar in underestimating summer values while those
for winter also often overestimated in the WBD simulation when very strong peaks occur (up to several 100 $\mu gm^{-3}$, e.g. for
Prague, Munich or Warsaw reaching almost 500 $\mu gm^{-3}$) which are not seen in the noWBD simulation. This suggests probably
a strong overestimation of the wind-blown dust emissions generating these peaks.

In Figures 13 and 14 the annual cycle of monthly mean concentrations for PM2.5 and PM10 are shown. All PM2.5 con-
centrations fluctuate with the same trend, having their highest values during winter and autumn seasons. The magnitude of the
difference between the modelled data WBD and noWBD and the observations is around 5-10 $\mu gm^{-3}$. Summer months are
underestimated while the inclusion of wind-blow dust reduces this negative bias. In winter the modelled values are overesti-
mated in Munich and Prague while underestimated in Berlin, Budapest and Warsaw. Depending on this, the inclusion of dust
emissions increases (e.g. Prague, Munich) or decreases (Vienna, Warsaw, Budapest) the model bias. In case of PM10, summer
values are underestimated in noWBD simulation by about 10-20 $\mu gm^{-3}$, while this underestimation is clearly reduced for the
WBD simulations to 0-10 $\mu gm^{-3}$. A different situation occurs in winter, when the noWBD model values underestimate the
measured data (by a similar magnitude as in summer), however, the inclusion of dust emissions increases model values such
that a positive model bias is generated. This is in line with the daily time-series seen above when strong peaks occur in the
WBD simulation being probably the main cause of these seasonal biases.

To gain more quantitative information on whether the inclusion of wbdust emissions reduced/enhanced the model biases, we
calculated several statistical measures presented below.

In the Tab. 1 and 2, the Pearson correlation coefficient, the Root Mean Squared Error (RMSE) and the Normalized Mean
Bias (NMB) were calculated for the daily mean concentrations of PM2.5 and PM10 in each city based on all values and on
seasonal selection. We calculated the statistics separately for WBD and noWBD ISORROPIA simulations.

The Pearson correlation measures the strength of the linear relationship between the modelled data and the observed one.
RMSE is the standard deviation of the residuals (prediction errors). It tells how concentrated the data is around the line of





best fit and lastly, finally, the Normalised Mean Bias (NMB) indicates the average deviation of the modelled values from the observed ones. These statistics are generated from all model-observation pairs from each station in a particular city.

The annual correlations of daily PM2.5 and PM10 with measurements are around 0.5-0.7 depending on the city, while the seasonal values are smallest for JJA around 0.2-0.4 and highest in DJF and MAM. An important result is that the correlations are much smaller for the WBD simulations, which indicates that the wind-blown dust emissions are poorly correlated with the real dust emissions that occurred. This is seen also for the RMSE, which has values between 5 and 20 $\mu$gm$^{-3}$ for PM2.5 and between 10-40 $\mu$gm$^{-3}$ for PM10 and evidently, the WBD values are higher. On a seasonal level, the lowest RMSE are encountered for JJA. In case of mean bias, annual values for PM2.5 are up to -0.2 for the noWBD simulations. In this case, the WBD brought improvement for some cities resulting in lower absolute NMB. This is especially due to JJA values where NMB improved for all cities. For PM10, annual NMB are also negative and reach -0.37. The WBD annual NMBs are in this case also lower for almost each city compared to the noWBD values. On seasonal levels, the improvement, i.e. lower mean biases are also evident.

In summary, by including wind-blown dust emissions, the correlation of the daily PM2.5/10 values decreased strongly, and the RMSE increased. However, the NMB improved for PM10 for almost all seasons and cities, while for PM2.5, the improvement occurred only for summer months.

### 3.3 Impact of WBD emissions on PM

In this section, the spatial distribution and the temporal evolution of the impact of dust emissions on PM2.5 and PM10 concentrations is presented (i.e. the $\Delta PM2.5$ and $\Delta PM10$ from Eq. 3 and 6). Starting with the temporal evolution, Fig. 15 and 16 represent the WBD impact to PM2.5 and PM10 concentrations for selected cities in central Europe.

The WBD impact to PM2.5 daily urban concentrations can reach values up to 30 $\mu$gm$^{-3}$, where the highest values are noticed in Berlin contributing up to 60 $\mu$gm$^{-3}$ to the total PM2.5 concentrations. The corresponding WBD impact to the PM10 concentrations are higher as expected and can reach values more than 80 $\mu$gm$^{-3}$, with Berlin representing again the highest extremes with values up to 200 $\mu$gm$^{-3}$. It is also clear that the highest impacts on PM are modelled during winter time in accordance with the annual cycle of emissions seen earlier.

To obtain spatial information on the WBD impact on PM, ,Fig. 17 depicts the seasonal averaged (2007-2016) dust impact on PM2.5 (left; $\Delta PM2.5$) and PM10 (right; $\Delta PM10$) concentrations above central Europe. The dust contribution to PM2.5 concentrations can reach values up to 12 $\mu$gm$^{-3}$ in DJF and about 8 $\mu$gm$^{-3}$ in other seasons while the highest impacts are modelled over Germany and over central Europe near large urban areas. In winter, a large part of the domain exhibits impact above 1 $\mu$gm$^{-3}$. The impact on PM10 is characterised with higher values up to 20 $\mu$gm$^{-3}$, mainly during DJF. while the spatial distribution is very similar to the PM2.5 impact being highest above western Europe (mainly Germany) with values above 2 $\mu$gm$^{-3}$ over other areas. The impacts seen are in line with the highest emissions calculated in Fig. 2.





### 3.3.1 Impact on PM components

As mentioned above, PM2.5 concentrations contain secondary constituents and in this section, we investigate how the presence

of wind-blow dust (FCRS and elements Ca, Fe, Mg, Mn, K, Na) would affect the concentrations of the anthropogenic secondary inorganic aerosols. Fig. 18 depicts the seasonally averaged WBD impact on PSO4, PNO3 and PNH4 for the ISORROPIA experiment.

Regarding sulphates, the strongest impacts occur during the winter reaching 0.1 $\mu gm^{-3}$ over parts of Germany and Poland. In other seasons the impact remains less than 0.05 $\mu gm^{-3}$ while it can be slightly negative in summer reaching -0.01 $\mu gm^{-3}$.

PNO3 is shown to be increased with the presence of WBD too with values up to 0.1 $\mu gm^{-3}$ in all seasons while most of the domain exhibit an increase above 0.01 $\mu gm^{-3}$. Finally, PNH4 is decreased above the entire domain with values often exceeding -0.05 $\mu gm^{-3}$ with peaking decreases around -0.1 $\mu gm^{-3}$ reached especially over the western part of the domain in winter.

The impact of WBD on secondary inorganic aerosol in the EQSAM experiments (Fig. 19) is evidently stronger in magnitude. The impact on PSO4 sometimes exceeds 0.1 $\mu gm^{-3}$ and also the negative impact over Italy is stronger. In case of nitrates the

impact also sometimes exceeds 0.1 $\mu gm^{-3}$ and a larger area is marked with increase above 0.05 $\mu gm^{-3}$ compared to the ISORROPIA. Finally, for ammonium, the decrease is larger than 0.01 $\mu gm^{-3}$ and can exceed 0.1 $\mu gm^{-3}$ being evidently stronger than in the ISORROPIA experiment.

The geographical distribution of the seasonally averaged impact does not provide information about the possible daily extremes of the impacts of WBD on secondary aerosol. Therefore we also plotted the temporal evolution of the daily averaged

change of PSO4, PNO3 and PNH4 concentrations due to WBD over the six selected urban areas.

Fig. 20 shows the WBD impact on the sulphates, while Fig. 21 and 22 shows the impact on nitrates and ammonium, respectively.

In contrast with the seasonal low impact of WBD to the PSO4 concentrations, daily extreme values show an impact up to 0.5 $\mu gm^{-3}$, while for some cities like Berlin even higher than 1 $\mu gm^{-3}$. These usually occur during the cold part of the year

in accordance with the spatial spatial results presented earlier. Daily WBD impact on nitrates is shown to be also higher than the seasonal one, with values reaching 1-1.5 $\mu gm^{-3}$. The WBD impact on ammonium seems to have a decreasing effect, with values ranging between -0.1 up to -0.5 $\mu gm^{-3}$, significantly higher than the seasonal ones as well.

## 4 Discussion and conclusions

This study looked at the potential long-term regional impact of dust emissions to PM2.5 and PM10 concentrations for the

period 2007-2016. The analysis focused on Central Europe and on big urban areas such as Berlin, Prague, Vienna, Munich, Budapest and Warsaw. The impact was also estimated for the secondary inorganic aerosol concentrations as constituents of PM2.5.

In our simulations, the annual average coarse and fine PM emitted averaged over the whole domain is about 1.5 and 0.5 $Mgkm^{-2}yr^{-1}$, so about 2 $Mgkm^{-2}yr^{-1}$ for the total PM10. This is by an order of higher value than calculated by Korcz et

al. (2008) for Europe. The dust emissions show significant temporal and spatial patterns. Our dust model computed 2 times





stronger emissions during winter period than in summer (for both the fine and coarse dust particles). Over high latitude areas, Bullard et al. (2016) reported strong winter dust emissions over areas where under dry conditions sublimation of snow (and eventually permafrost) occurs and the soil is more prone to saltation, while during summer, the soil is generally more moist reducing the saltation potential of soil particles. In the dust model used in this study, the three most important parameter

affecting the dust emissions is the near-surface wind speed, snow equivalent water and soil moisture. The reason for much higher winter wbdust emissions can be in 1) much higher winter windspeeds modelled compared to summer ones, lower soil moisture during winter months and underestimation of snow cover which prevents dust events. We saw that our driving model (WRF) produced much higher winds than the measured ones and this positive bias is largest during winter. This strong overestimation is a known feature of the BouLac PBL scheme used in this study and others reported similar overestimation of

wind speed (e.g. Tyagi et al., 2018; Zhang et al., 2021). As dust emissions scale non-linearly with wind speed that are above a threshold (Leung et al., 2022) this raises the potential to overestimate dust emissions if winds are overestimated.

Regarding the modelled soil moisture, it is comparable to observed values in winter and somewhat higher during other months than those measured. This means that the strong winter emissions are probably due to high wind-speeds in WRF. The last factor potentially playing a role is the snow cover which was not evaluated in this study, but the modelled precipitation

exhibited some underestimation in winter which might result in reduced snow in our simulations (even though temperature is underestimated in winter).

Apart from the clear annual cycle, the calculated emissions show a diurnal cycle too. Daytime emissions are usually 50-100% higher than nighttime ones. The reason for this is most probably due to the well-known cycle of wind-speed with maxima occurring during noon time (Huszar et al., 2018, 2020) and similar diurnal behaviour of dust emissions were seen in

other studies too (e.g. Klose et al., 2012).

The daily timeseries of dust emissions provide some hint on their distribution: while most of the days, the coarse mode emissions remain low (lower than 10-20 $\mathrm{gs^{-1}grd^{-1}}$, which is about 100-200 $\mathrm{mgs^{-1}km^{-2}}$), on selected day the emissions peak at much higher values (100-200 of $\mathrm{gs^{-1}grd^{-1}}$, i.e. 1000-2000 $\mathrm{mgs^{-1}km^{-2}}$). The same is true for the fine mode dust. This points to the fact already mentioned that the dust emissions respond non-linearly to wind speeds (more specifically to

friction velocity, see Eq. 2) above a certain threshold. As wind speeds are overestimated in the driving meteorological model (WRF), probably the emissions are also overestimated, or at least these strong peaks are not realistic.

One interesting feature is evident from the modelled geographical distribution of seasonal WBD emissions. Besides large rural areas emitting dust, the largest dust sources are concentrated near large urban areas: the Ruhr area in Germany, the highly populated Benelux states, and also other large cities like Berlin, Prague, Budapest etc. One has to be very cautious

in interpreting this result: dust is potentially emitted only over landuse categories representing potentially bare soil (if other circumstances are met), i.e. crops, shrubs, grass land, tundra and desert. "Urban" landuse is however not treated with dust emission potential. On the other hand, urban areas are characterised with low vegetation and thus leaf-area-index (LAI). Indeed, in the used LAI data (MODIS), cities have near zero LAI during most of the year. As landuse used for wbdust module (and also for CAMx dry deposition) is represented as fractional landuse (based on CORINE data), many of the 9 km gridboxes covering

urban areas are partly covered with bare soil and partly urban landuse. If the the LAI value is too low for such areas, the dust





emission can occur in the model. This was the case for densely populated areas e.g. over the mentioned Rurh area. In the case of large cities, as e.g. Berlin, dust emissions are concentrated near the edges of the city, where gridboxes share both urban landuse and bair soil. It is difficult to evaluate how realistic such "near"-urban dust emissions are, but this has to be treated as a cautious note on providing consistent input data for landuse and other land-related parameters like LAI.

Regarding the PM concentrations, the background noWBD case showed a reasonable model performance with typical correlations for PM2.5 and PM10 achieve in other modelling studies for Europe (e.g. Lecœur et al., 2013; Tsyro et al., 2022). Lowest correlations are computed for summer period while winter ones are usually the highest. This can be explained by the more stable weather conditions during DJF which are better resolved than summer weather often marked with highly variable convective environment (Huszar et al., 2016). The PM values are underestimated in summer and overestimated in winter which

is probably due to too strong vertical transport in summer and too low in winter, but may be connected also to deficiencies in the monthly profiles used for annual emissions (Huszar et al., 2018, 2020). An important goal of the model validation was to evaluate whether the inclusion of WBD emission improves model performance. This turned out to be true for summer biases, which were reduced by adding the dust load. However, the winter which was already marked with negative bias, is modelled with even higher bias if dust is considered. Also the correlations decreased significantly if wind-blown dust is included in our

simulations. This can be explained by the strong peaks in the impact on PM values which are a result of strong emission peaks seen in the daily time series of FCRS and CCRS emissions. The modelled urban PM peaks are often much higher (often by a factor of 5 or even more) than measurements and thus can strongly reduce the correlation with the observed values.

Similar to dust emissions, the WBD impact on overall PM concentrations is high near big urban centres (over Germany and the Benelux states; reaching 15-20 $\mu gm^{-3}$), but large rural contributions are modelled exceeding 2 and 5 $\mu gm^{-3}$ too for

PM2.5 and PM10, respectively. The contributions are largest, in accordance with the largest emissions, during winter. These seasonally averaged impacts are however strongly exceeded by the daily average values which can be higher by 1 order of magnitude reaching 100 $\mu gm^{-3}$ for some cities. However, as already said, these extreme peaks are probably overestimated (due to too strong winds in WRF).

Vautard et al. (2005) calculated the summer and autumn wind-blow dust contribution to PM due to European local sources

and found around 1-2 $\mu gm^{-3}$ contribution to PM10 over central Europe which is about 2 times less than in our simulations.

Apart from the impact on the overall PM2.5/10 concentrations, our study quantified the long-term impact on the aerosol secondary components, namely the secondary inorganic aerosol (SIA) components. In seasonal average, the impact are rather small, around up to 0.1 $\mu gm^{-3}$ increase for PSO4 (mainly during winter) and PNO3 (all year round). For PNH4, we modelled decreases of similar absolute magnitude. Much higher impact is however calculated for specific days as daily means. They

can reach up to 0.5-1 $\mu gm^{-3}$ increase for sulphates (maximum increase over Berlin in 2009 exceeding 1 $\mu gm^{-3}$) and similar increases for nitrates. During winter 2008-2009, nitrates occasionally even decreased up to 0.2-0.3 $\mu gm^{-3}$. Ammonium decreased due to dust by up to -1 $\mu gm^{-3}$ on selected days. These decreases occurred mainly during winter days.

The explanation of the above presented SIA modifications can be explained by two types of processes: one is the heterogeneous oxidation from SO2 and N2O5 on the surface of dust particles (Wang et al., 2012; Zheng et al., 2015) and the other is

the catalytic oxidation enhancement by dust elements in cloud water via influencing cloud pH and thus the aqueous chemistry





of $SO_2$, $HNO_3$ and $NH_3$. Indeed, if we consider the impact of total SIA, we see that they increased (the increases of sulphates and nitrates overweight the decrease of ammonium). This is in line with the expectation and with previous studies dealing with the impact of dust on secondary aerosol formation (Malaguti et al., 2015).

Concretely, the increases of sulphates due to presence of dust particles were modelled by Wang et al. (2012) (increases by
about 1 $\mu gm^{-3}$ during a strong dust event in China) who attributed it to dust surface heterogeneous chemistry. Also Kakavas and Pandis (2021) modelled increases of sulphates over Europe due to dust. Our findings, at least qualitatively, are also in line with the recent findings of Wang et al. (2022) who argued that on the "dust surface, heterogeneous drivers are more efficient than surface-adsorbed oxidants in the conversion of $SO_2$, particularly during nighttime".

Regarding the impact on nitrates, the increases are consistent with an earlier study of Fairlie et al. (2010) who found that
nitrates associate with dust and result in volatilisation. The increase of nitrates can be explained also by the formation of deliquescent salts (e.g., through the reaction of crustal cations in dust with $NO^{3-}$ ions) as argued also by Wang et al. (2012). This can potentially lead to even some over-prediction of nitrates which requires the revisiting the chemical composition of dust (Karydis et al., 2011).

Finally, the ammonium response to dust is tightly connected to the response of sulphates and nitrates. As we saw that nitrates
easily associate with dust (via reaction with dust crustal anions like $Ca^{2+}$) this means that less nitrates is available to react with ammonium leading to more ammonia remaining in gas phase (Fairlie et al., 2010). In other words, $NH4^+$ is replaced by dust contained cations (Wang et al., 2012). This latest author and others (e.g. Malaguti et al., 2015) note that ammonium could also increase due to dust presence as a result of more sulphates forming on dust surfaces. However, in our simulations this is evidently offset by the above mentioned replacement of ammonium by crustal cations.

In our experiments, the impact on SIA is clearly stronger using the EQSAM equilibrium model, although the differences are not large and the overall impact on PM is not affected too much. The reason for stronger sulphate and nitrate formation in EQSAM is probably due to the fact, that in EQSAM, the cloud pH is influenced with three cations (Mg+, Ca++ and K+) while in ISORROPIA, it is only Calcium. This also explains the stronger decrease of ammonium in EQSAM being replaced by more cations.

An exception of the above mentioned behaviour for nitrates is the winter 2009 decrease (by about 0.2 $\mu gm^{-3}$) seen for all analysed cities. This period is not characterised by exceptional dust emissions nor extreme PM values (based on our results). On the other hand, during this period, the dust impact on PSO4 is relatively large while the impact on ammonium is very small. Probably thus ammonium was neutralising preferably sulphates instead of nitrates causing their reduction.

Summing up the results, we showed that the long-term impact of local wind-blown dust emission in Europe can significantly
enhance urban PM levels, especially during extreme events rather then in seasonal averages. However, our calculations are probably overestimating dust emissions due to very strong winds in the driving model. We also showed that apart from the total aerosol load, dust impacts also the secondary inorganic fraction of PM which can significantly increase during selected days.

We have to note also that the uncertainties related to different inputs used for the study cannot be judged here. We already mentioned that the landuse and the LAI input can co-act (bare soil vs. low/high LAI) differently depending on the choice of
these data. This caused for example that in our simulations one of the highest dust emissions are located around urban areas. We



also used some default values for dust composition based on a study (Karydis et al., 2011) which measured this composition over a different geographic area. Lastly, we used only one driving model and one model for wind-blown dust emissions so the model uncertainty also cannot be addressed. The future goal should be thus to focus on the sensitivities of wind-blown dust loads to different input data and methods to obtain a more robust long-term estimate of their emissions and impact on PM and

their secondary components.

*Data availability.* CAMx version 7.10 is available at http://camx-wp.azurewebsites.net/download/source (last access: 30 November 2022; CAMx, 2020; Ramboll et al., 2020). WRF version 4.0 ca be downloaded from https://www2.mmm.ucar.edu/wrf/src/WRFV4.0.TAR.gz (last access 30 November 2022; WRF (2022)). The source code of the WBDUST model can be downloaded from the CAMx "Support Software" page: https://www.camx.com/download/support-software/ (last access 30 November 2022; WBDUST (2022)). The LAI data used

in WBDUST is obtained from http://globalchange.bnu.edu.cn/research/laiv6 (Last access 30 November 2022; (Yuan et al., 2011)). The complete model configuration and all the simulated data (three-dimensional hourly data) used for the analysis are stored at the Department of Atmospheric Physics, Charles University data storage facilities (about 5 TB), and are available upon request from the main author. The observational data from the AirBase database can be obtained from https://discomap.eea.europa.eu/map/fme/AirQualityExport.htm (EEA, 2021). The data from the Czech Hydrometeorological Institute AIM network can be obtained upon request from the authors.

*Author contributions.* ML and PH conceptualized and designed the experiments and wrote the majority of the text, PH conducted the CAMx simulation, JK performed the WRF experiments, ML, LB, APPP contributed to the analysis of the results and OV helped with obtaining the observational data and writing the text.

*Competing interests.* No competing interests are present.

*Acknowledgements.* This work has been supported by the Czech Technological Agency (TACR) grant No.SS02030031 ARAMIS (Air Qual-

ity Research Assessment and Monitoring Integrated System) and Charles University Grant Agency (GAUK) project no. 298822. It has been partly funded also by the Austrian Climate and Energy Funds via project ACRP11-KR18AC0K14686 and the Charles University SVV 260581 project. We also further acknowledge the TNO-MACC-III emissions dataset provided by the Copernicus Monitoring Service, the compiled air quality station data provided by the European Environmental Agency, the ERA-Interim reanalysis provided by the European Centre for Medium-Range Weather Forecast, the MODIS leaf-area-data provided by Land-Atmosphere Interaction Research Group at Sun

Yatsen University. We also thank the Czech Hydrometeorological Institute providing the AIM data.





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

**Figure 1.** The input data for the clay fraction in % (top) and the vegetation factor for January and July (middle and bottom, respectively) based on MODIS 2010 LAI data

Zheng, B., Zhang, Q., Zhang, Y., He, K. B., Wang, K., Zheng, G. J., Duan, F. K., Ma, Y. L., and Kimoto, T.: Heterogeneous chemistry: a mechanism missing in current models to explain secondary inorganic aerosol formation during the January 2013 haze episode in North China, Atmos. Chem. Phys., 15, 2031–2049, https://doi.org/10.5194/acp-15-2031-2015, 2015.

Zittis, G., Almazroui, M., Alpert, P., Ciais, P., Cramer, W., Dahdal, Y., et al.: Climate change and weather extremes in the Eastern Mediterranean and Middle East. Rev. Geophys. 60, e2021RG000762. doi:10.1029/2021RG000762, 2022.

Zwaaftink G., C. D., Aas, W., Eckhardt, S., Evangeliou, N., Hamer, P., Johnsrud, M., Kylling, A., Platt, S. M., Stebel, K., Uggerud, H., and Yttri, K. E.: What caused a record high PM10 episode in northern Europe in October 2020?, Atmos. Chem. Phys., 22, 3789–3810, https://doi.org/10.5194/acp-22-3789-2022, 2022.

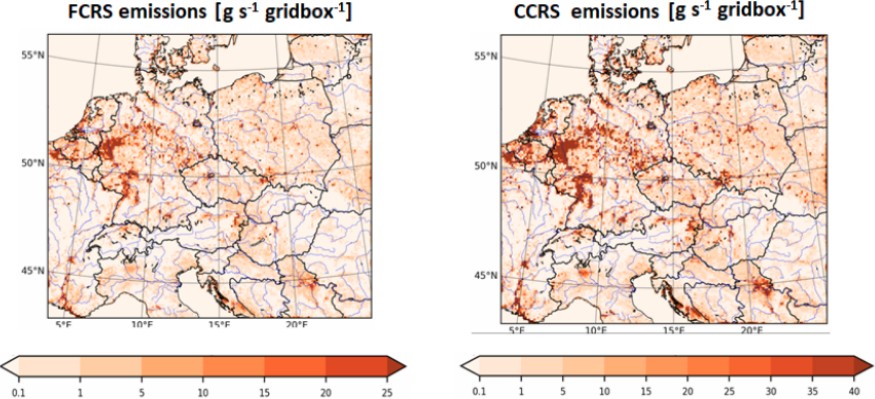

**Figure 2.** Average seasonal WBD emission fluxes of Fine Crustal material (FCRS; left) and of Coarse Crustal material (CCRS; right) above central Europe in DJF for 2007-2016 period in $\mathrm{gs^{-1}gridbox^{-1}}$. Note, the colorbars are different.

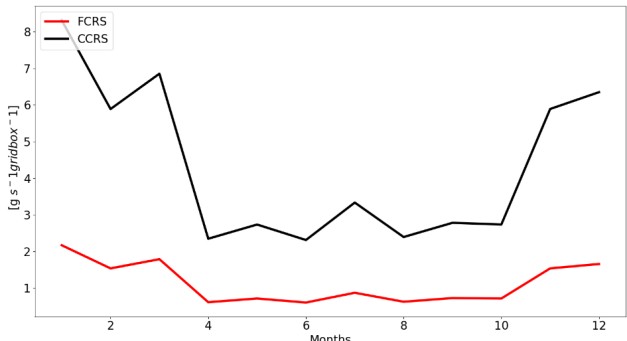

**Figure 3.** Domain-averaged annual cycle of monthly averages of FCRS and CCRS WBD emission fluxes for 2007-2016 in $\mathrm{gs^{-1}gridbox^{-1}}$.





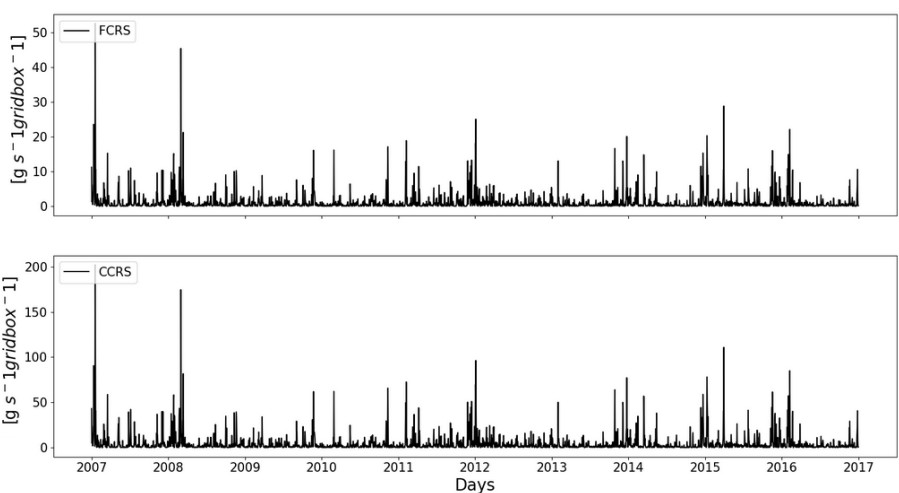

**Figure 4.** Domain-averaged daily WBD emission fluxes of FCRS (top) and CCRS (bottom) for 2007-2016 in $\mathrm{gs^{-1}gridbox^{-1}}$.

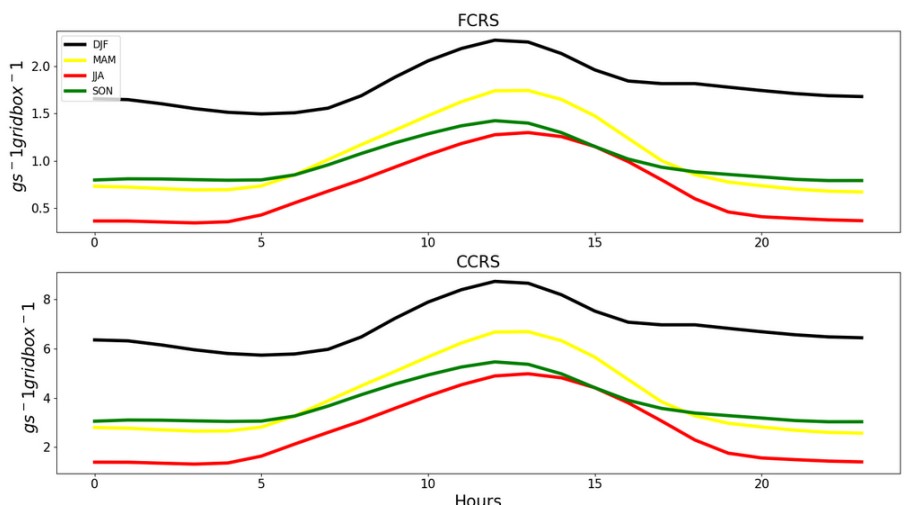

**Figure 5.** Domain-averaged diurnal cycle of hourly FCRS and CCRS WBD emission fluxes for different seasons in 2007-2016. Units in $\mathrm{gs^{-1}gridbox^{-1}}$.

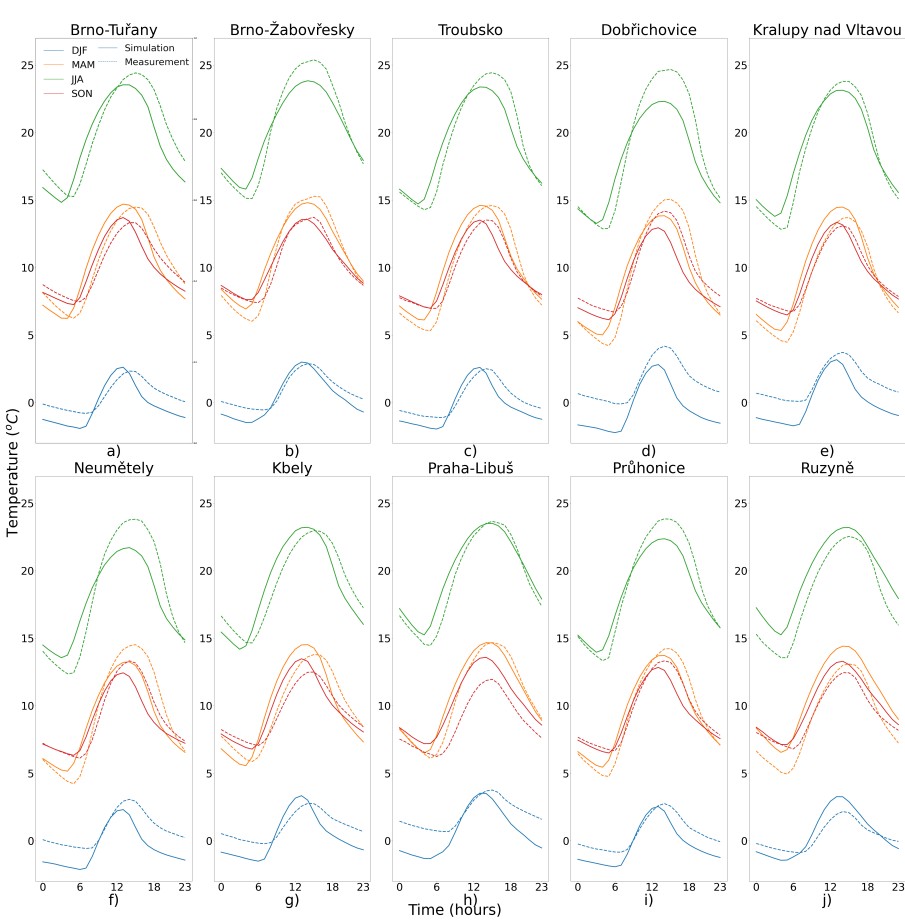

**Figure 6.** Comparison of modelled temperature diurnal profiles (solid) with measurements (dashed) from 10 Czech stations averaged over different seasons for the 2007-2016 period. Units in C°

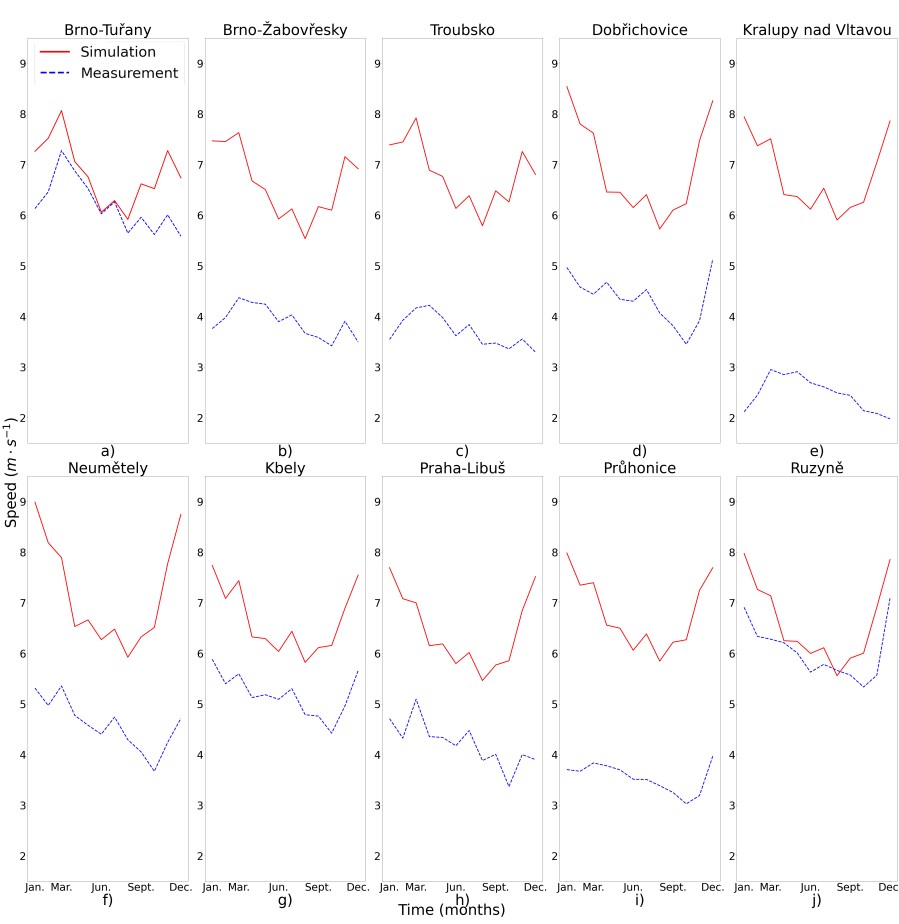

**Figure 7.** Comparison of modelled annual cycle of the monthly mean of maximum daily wind-speeds (red solid) with measurements (blue dashed) from 10 Czech stations averaged over the 2007-2016 period. Units in $\mathrm{ms}^{-1}$





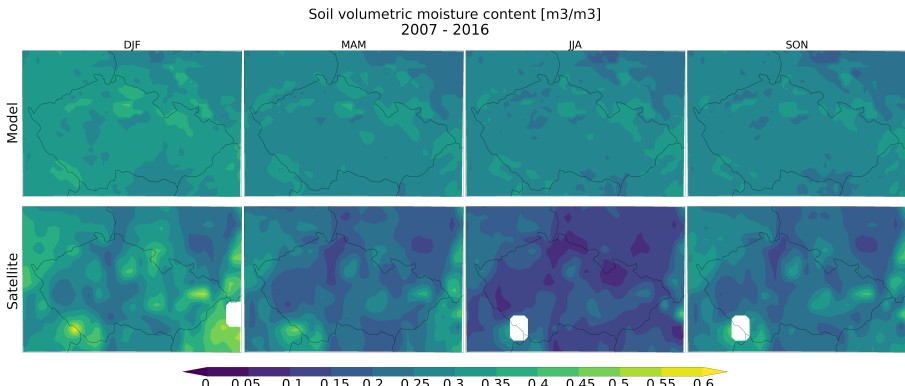

**Figure 8.** Comparison of modelled seasonal volumetric soil moisture (upper row) with the ESA CCI soil moisture data (lower row) for the area of Czech republic. Data averaged over 2007-2016. Units in $\mathrm{m^3 m^{-3}}$.

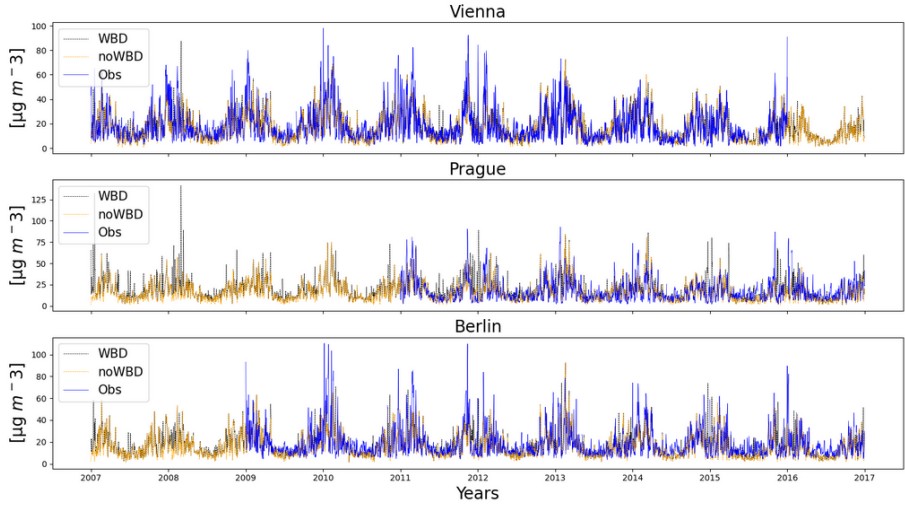

**Figure 9.** PM2.5 daily averaged concentrations of WBD (black dashed), noWBD (orange dashed) and Airbase dataset (blue solid) for 2007-2016 (Vienna, Prague, Berlin). Units in $\mathrm{\mu g m^{-3}}$.



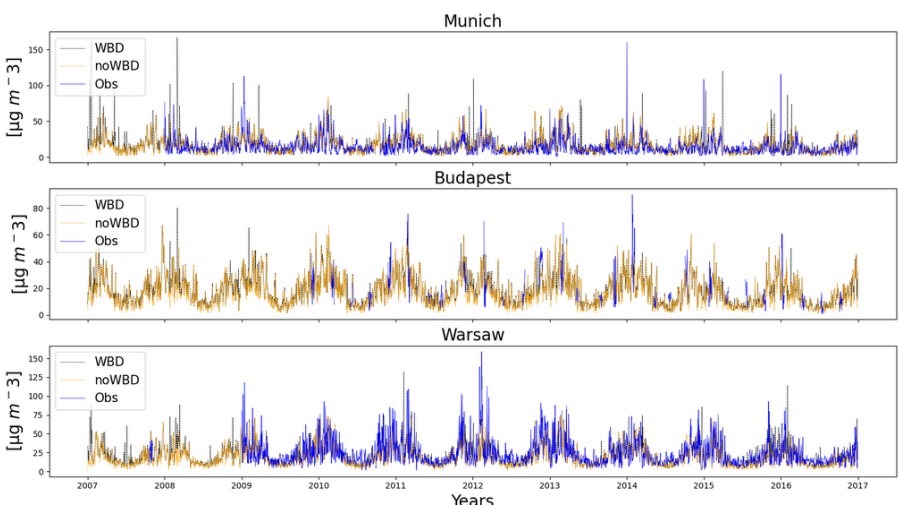

**Figure 10.** Same as Fig 11 but for Munich, Budapest, Warsaw.

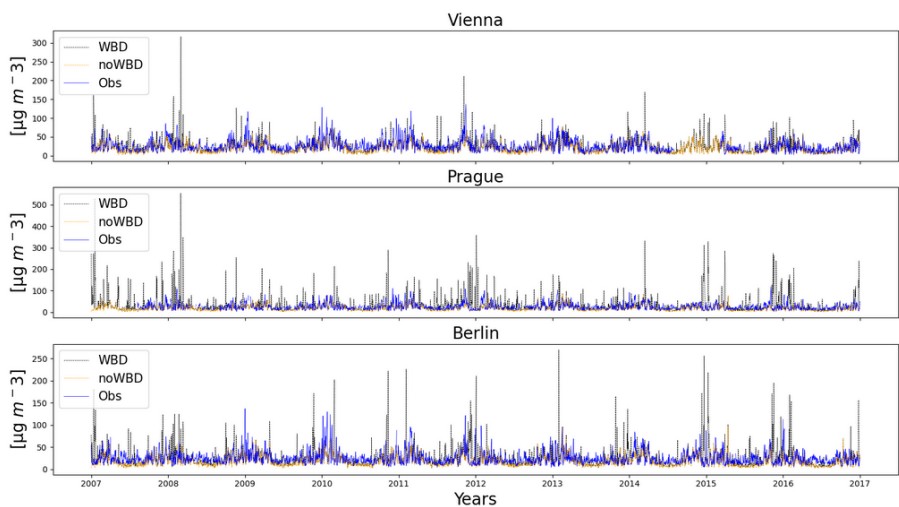

**Figure 11.** PM10 daily averaged concentrations of WBD (black dashed), noWBD (orange dashed) and Airbase dataset (blue solid) for 2007-2016 (Vienna, Prague, Berlin). Units in $\mu gm^{-3}$.




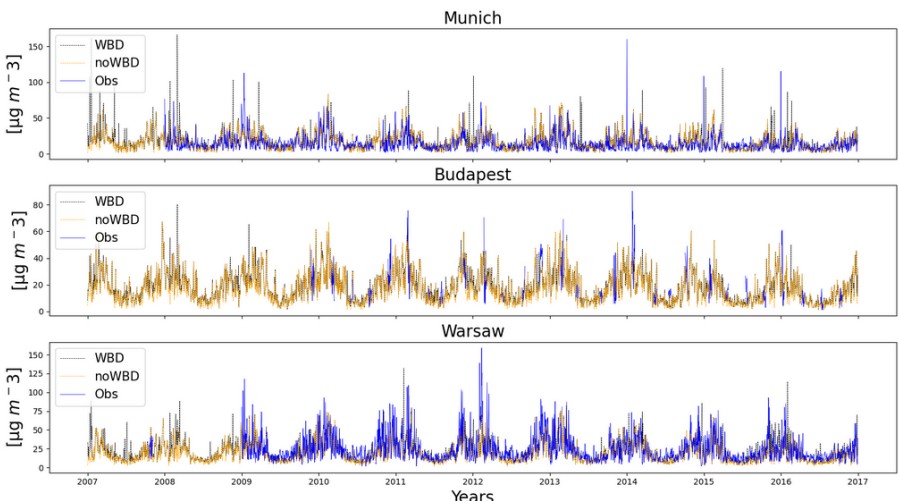

**Figure 12.** Same as Fig 11 but for Munich, Budapest, Warsaw.

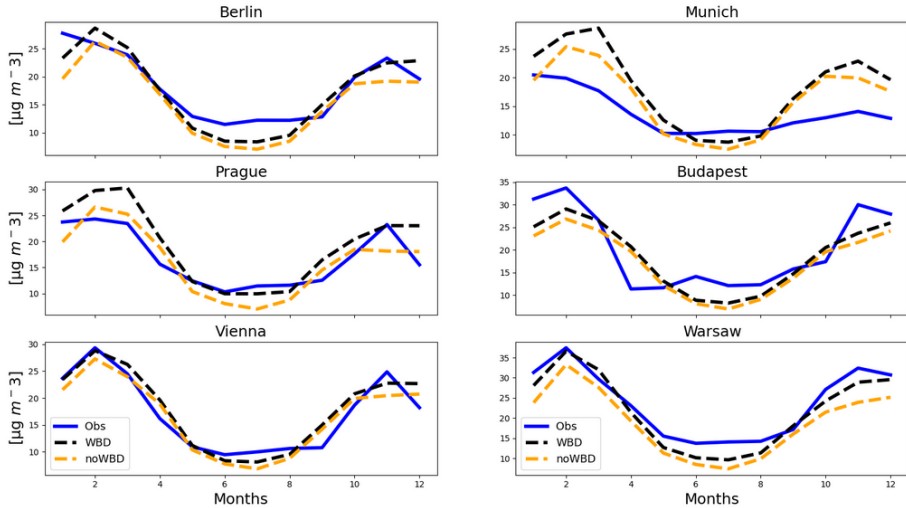

**Figure 13.** Annual cycle of monthly PM2.5 concentrations of WBD (blue dashed), noWBD (orange dashed) simulations and Airbase dataset (blue solid) for 2007-2016.



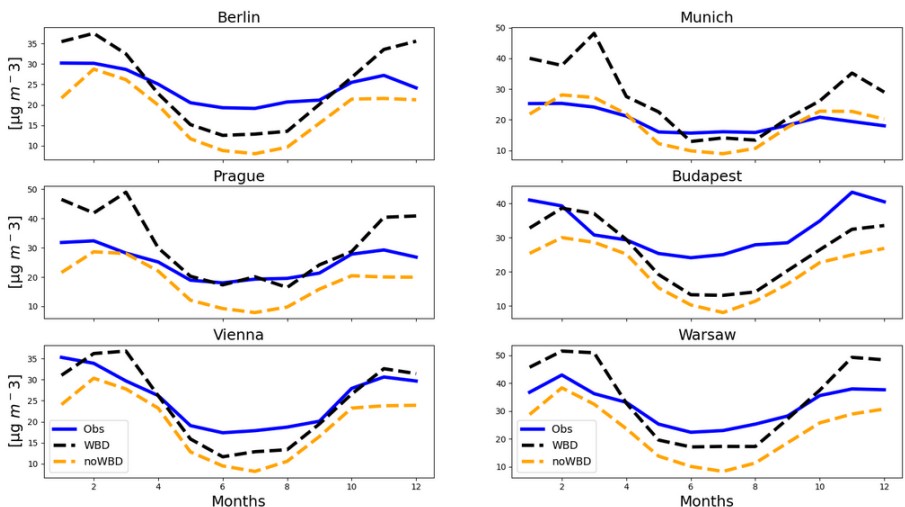

**Figure 14.** Annual cycle of monthly PM10 concentrations of WBD (blue dashed), noWBD (orange dashed) simulations and Airbase dataset (blue solid) for 2007-2016.



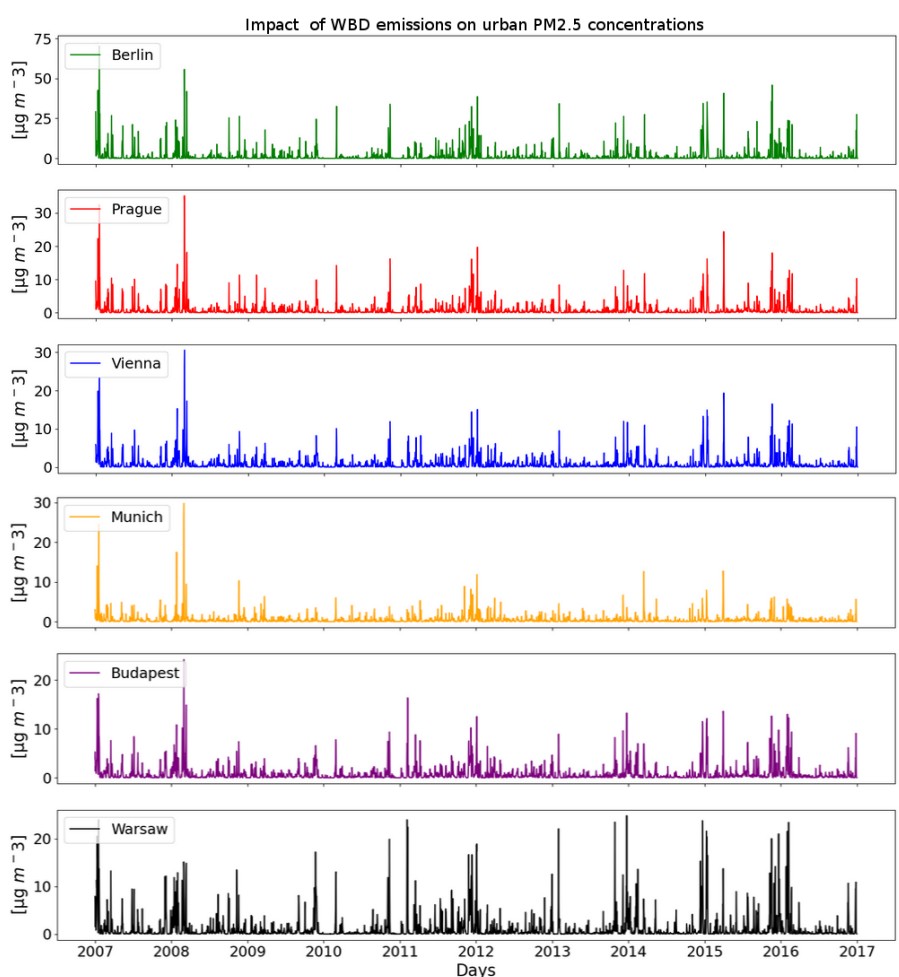

**Figure 15.** Daily averaged impact of wind-blown dust emissions on PM2.5 concentrations in $\mu g m^{-3}$ for 2007-2016.



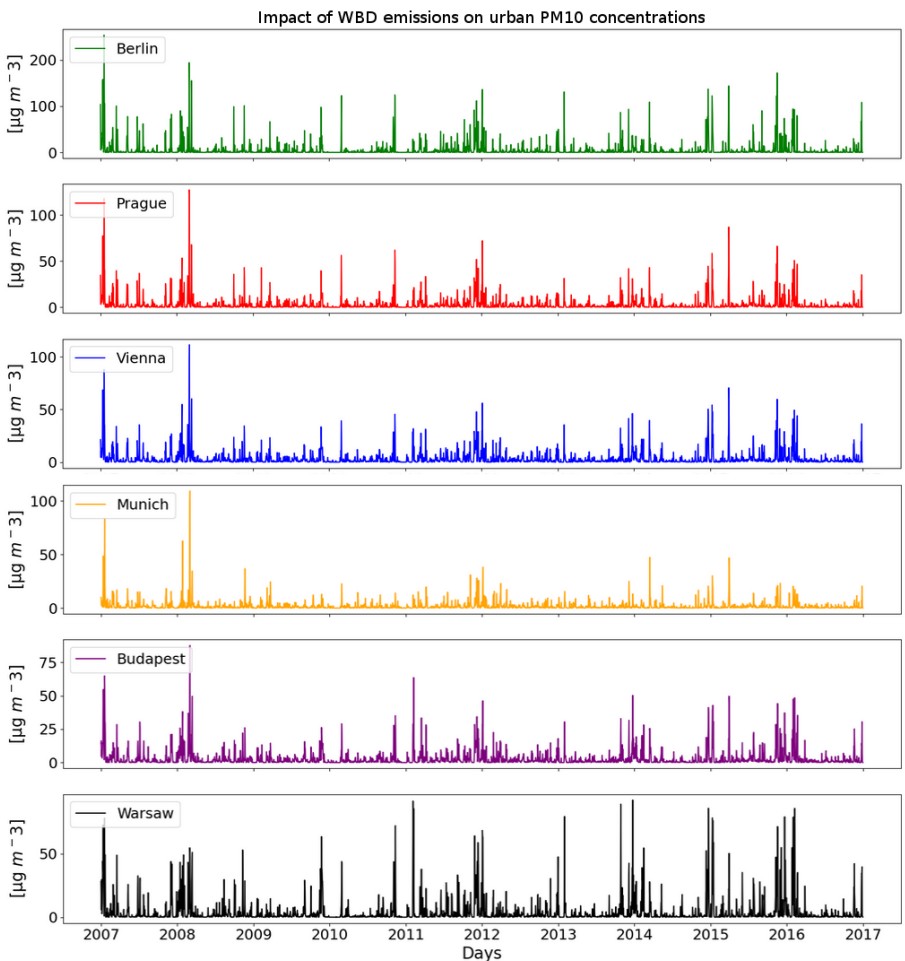

**Figure 16.** Daily averaged impact of wind-blown dust emissions on PM10 concentrations in $\mu g m^{-3}$ for 2007-2016.





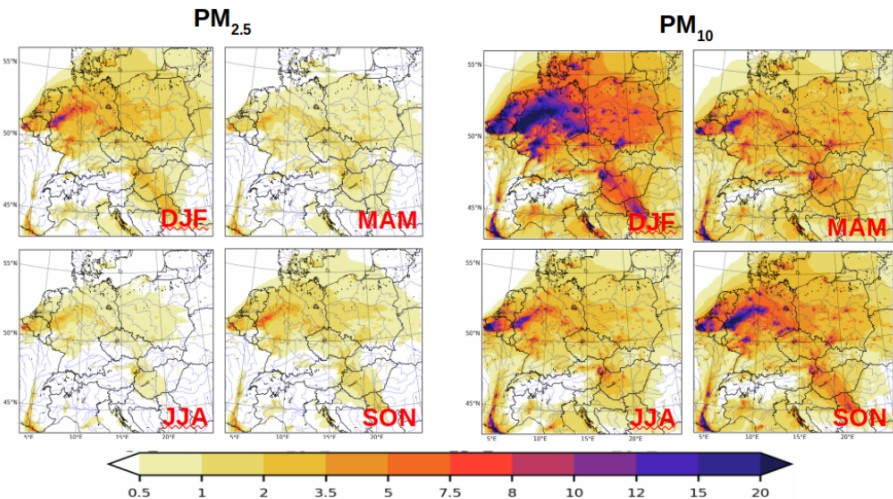

**Figure 17.** Seasonally averaged impact of WBD emissions to PM2.5 (left) and PM10 (right) concentrations in μgm$^{-3}$ for 2007-2016.

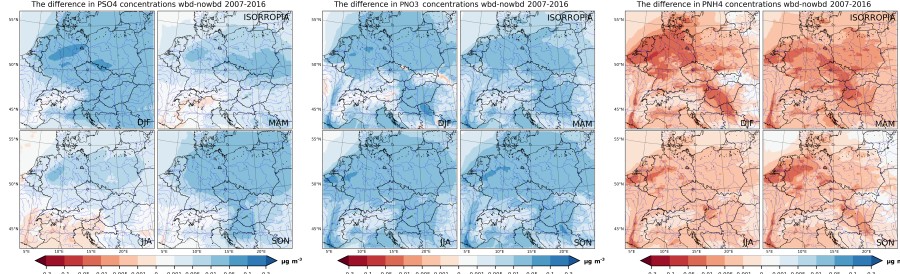

**Figure 18.** The WBD emission impact on secondary inorganic aerosol concentrations (PSO4, PNO3 and PNH4) with ISORROPIA, seasonally averaged, in μgm$^{-3}$ for 2007-2016.

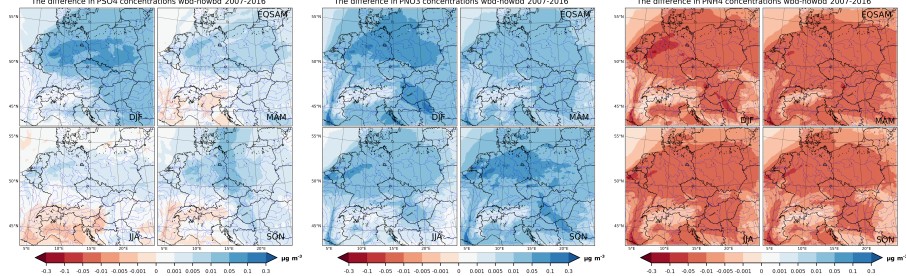

**Figure 19.** The WBD emission impact on secondary inorganic aerosol concentrations (PSO4, PNO3 and PNH4) with EQSAM, seasonally averaged, in μgm$^{-3}$ for 2007-2016.



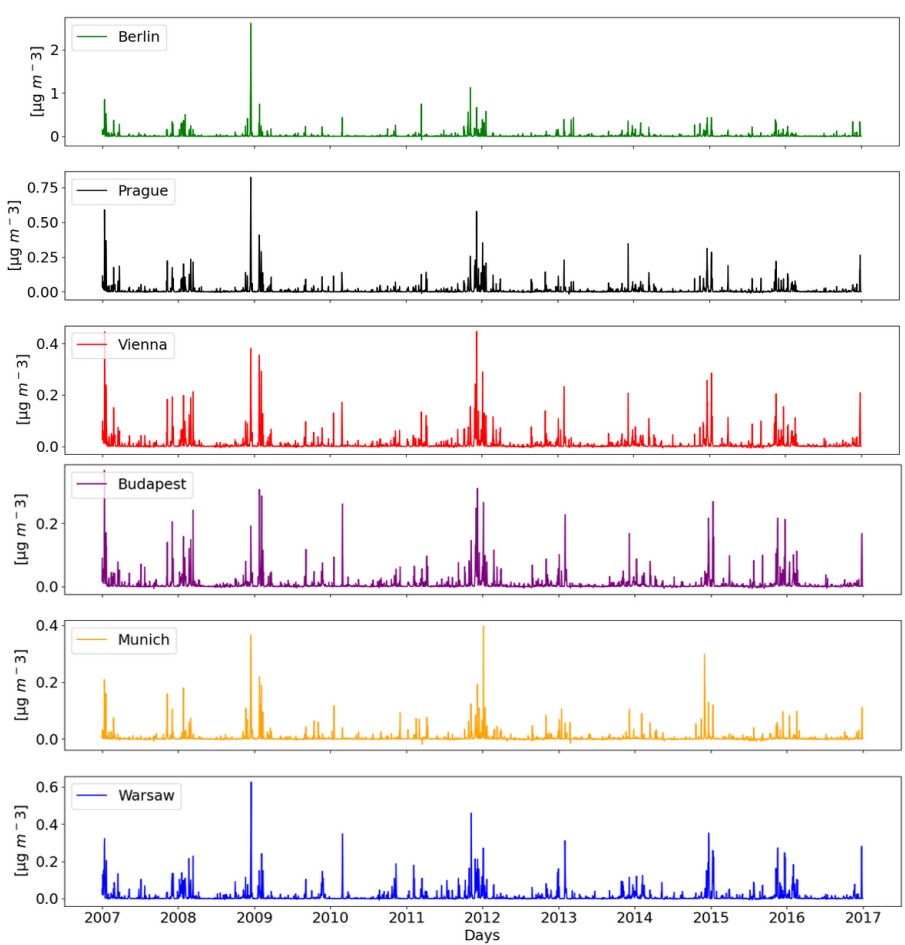

**Figure 20.** The long-term WBD impact on PSO4 concentrations in ISORROPIA, daily averaged for 2007-2016. Units in $\mu g\,m^{-3}$.



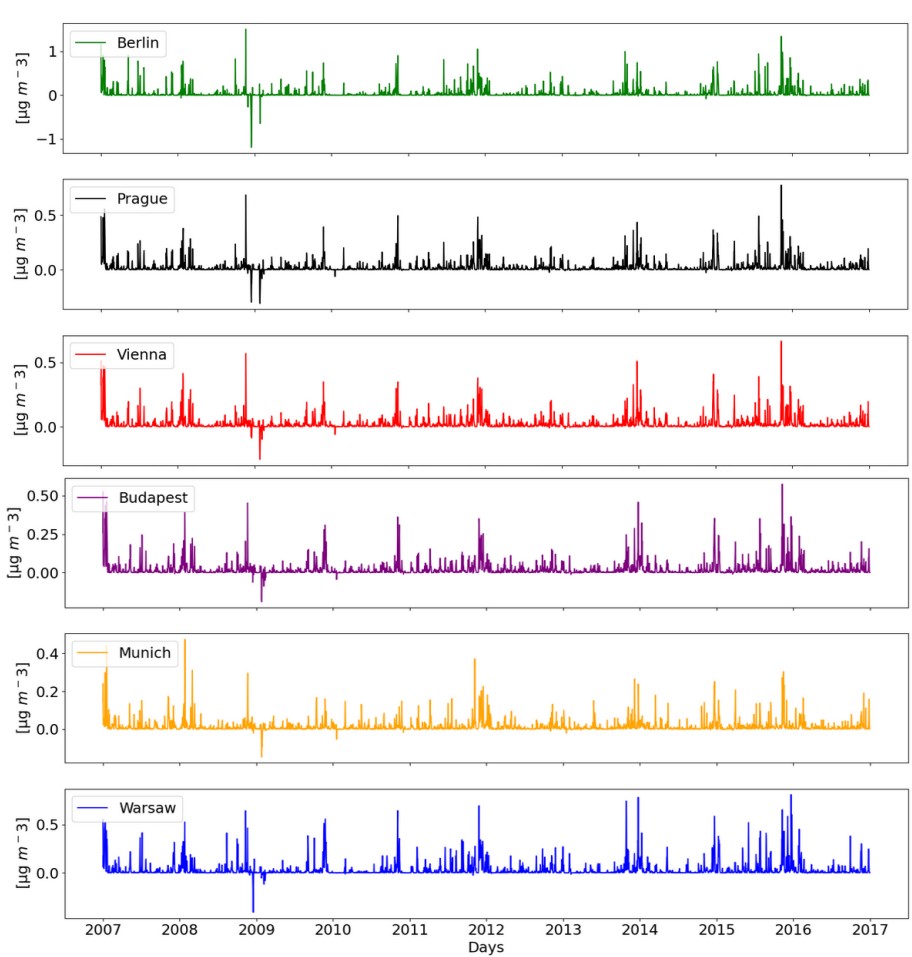

**Figure 21.** The long-term WBD impact on PNO3 concentrations in ISORROPIA, daily averaged for 2007-2016. Units in $\mu gm^{-3}$.



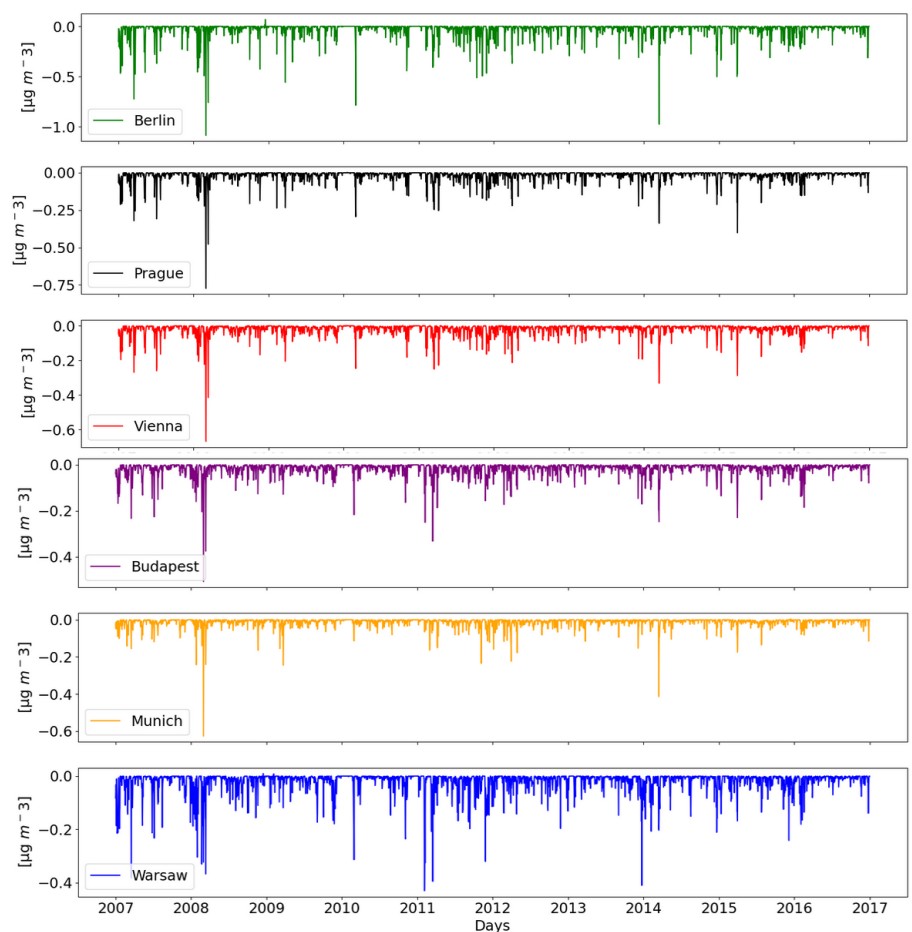

**Figure 22.** The long-term WBD impact on PNH4 concentrations in ISORROPIA, daily averaged for 2007-2016. Units in $\mu g m^{-3}$.





**Table 1.** Annual and seasonal statistical measures (Pearson correlation, RMSE, NMB) for PM2.5 for both WBD and noWBD ISORROPIA simulations calculated from the daily averages.

| Cities | PM2.5 | Pearson correlation | | RMSE [$\mu g m^{-3}$] | | NMB | |
|---|---|---|---|---|---|---|---|
| | | WBD | noWBD | WBD | noWBD | WBD | noWBD |
| Vienna | Annual | 0.6901 | 0.7146 | 9.6728 | 9.2239 | 0.0590 | -0.0168 |
| | DJF | 0.6123 | 0.6628 | 12.7105 | 12.0459 | 0.0338 | -0.0374 |
| | MAM | 0.6448 | 0.6852 | 9.5338 | 8.7539 | 0.1260 | 0.0539 |
| | JJA | 0.2534 | 0.3279 | 5.3553 | 5.1018 | -0.1008 | -0.1861 |
| | SON | -0.1031 | -0.1056 | 9.3356 | 9.3380 | 0.1069 | 0.0261 |
| Prague | Annual | 0.4778 | 0.6897 | 12.8227 | 9.4919 | 0.1088 | -0.0748 |
| | DJF | 0.3349 | 0.6860 | 17.9578 | 12.5324 | 0.1639 | -0.0674 |
| | MAM | 0.6195 | 0.7670 | 11.403 | 7.9885 | 0.1977 | 0.0405 |
| | JJA | -0.0557 | 0.3106 | 6.8719 | 6.1530 | -0.1450 | -0.3323 |
| | SON | 0.4001 | 0.6020 | 12.8983 | 10.2484 | 0.1168 | -0.0370 |
| Berlin | Annual | 0.6291 | 0.5342 | 10.7016 | 9.3405 | -0.0340 | -0.1321 |
| | DJF | 0.4985 | 0.6772 | 16.1053 | 13.5632 | 0.0242 | -0.1036 |
| | MAM | 0.7176 | 0.7615 | 8.0587 | 7.4442 | -0.0127 | -0.0694 |
| | JJA | 0.1372 | 0.3264 | 5.9892 | 6.0888 | -0.2928 | -0.3792 |
| | SON | 0.6232 | 0.7338 | 10.0036 | 8.5933 | 0.0402 | -0.0676 |
| Munich | Annual | 0.4612 | 0.6236 | 12.9249 | 9.6508 | 0.2983 | 0.1740 |
| | DJF | 0.4322 | 0.5893 | 17.1053 | 13.9096 | 0.3125 | 0.1935 |
| | MAM | 0.4533 | 0.7364 | 14.3441 | 7.8727 | 0.4240 | 0.2427 |
| | JJA | 0.1882 | 0.2981 | 5.6866 | 5.4625 | -0.1405 | -0.2118 |
| | SON | 0.4907 | 0.6378 | 11.6281 | 9.3698 | 0.4971 | 0.3857 |
| Budapest | Annual | 0.6893 | 0.7269 | 10.7620 | 10.5346 | -0.1001 | -0.1690 |
| | DJF | 0.6386 | 0.7330 | 15.6802 | 15.3078 | -0.1673 | -0.2432 |
| | MAM | 0.6209 | 0.6372 | 9.6156 | 9.4123 | 0.1270 | 0.0683 |
| | JJA | 0.2858 | 0.4302 | 7.3253 | 7.4711 | -0.2976 | -0.3852 |
| | SON | 0.6965 | 0.7488 | 10.0019 | 9.6213 | -0.0883 | -0.1473 |
| Warsaw | Annual | 0.5697 | 0.6999 | 14.4706 | 12.9026 | -0.0922 | -0.2027 |
| | DJF | 0.3858 | 0.5844 | 20.6196 | 18.0144 | -0.0556 | -0.1545 |
| | MAM | 0.5601 | 0.6769 | 12.5453 | 11.0780 | -0.0504 | -0.1512 |
| | JJA | 0.1074 | 0.2841 | 7.4900 | 7.7487 | -0.2877 | -0.3993 |
| | SON | 0.5116 | 0.6932 | 14.1531 | 12.6325 | -0.0691 | -0.2021 |





**Table 2.** Annual and seasonal statistical measures (Pearson correlation, RMSE, NMB) for PM10 for both the WBD and noWBD ISORROPIA simulations calculated from the daily averages.

| Cities | PM10 | Pearson correlation | | RMSE [$\mu g m^{-3}$] | | NMB | |
|---|---|---|---|---|---|---|---|
| | | WBD | noWBD | WBD | noWBD | WBD | noWBD |
| Vienna | Annual | 0.3836 | 0.6766 | 18.7462 | 13.1320 | -0.0368 | -0.2334 |
| | DJF | 0.2464 | 0.6456 | 23.9934 | 17.2384 | 0.0002 | -0.1995 |
| | MAM | 0.3113 | 0.6709 | 20.0530 | 10.8410 | 0.0542 | -0.1494 |
| | JJA | 0.1730 | 0.4334 | 11.8061 | 10.9038 | -0.2901 | -0.4682 |
| | SON | 0.4095 | 0.6471 | 16.9706 | 12.7993 | -0.0028 | -0.2011 |
| Prague | Annual | -0.0221 | 0.6734 | 38.0248 | 13.1536 | 0.2422 | -0.2729 |
| | DJF | -0.1746 | 0.6618 | 51.8210 | 16.4792 | 0.3954 | -0.2156 |
| | MAM | 0.1036 | 0.7397 | 36.9492 | 9.7224 | 0.3559 | -0.1433 |
| | JJA | -0.2398 | 0.3799 | 19.5380 | 12.1537 | -0.0769 | -0.5414 |
| | SON | -0.0871 | 0.6478 | 35.9450 | 13.2200 | 0.1848 | -0.2749 |
| Berlin | Annual | 0.1914 | 0.6675 | 23.2173 | 12.4793 | 0.0210 | -0.2660 |
| | DJF | 0.0116 | 0.6662 | 35.7968 | 15.2236 | 0.2917 | -0.1479 |
| | MAM | 0.4385 | 0.6820 | 14.6269 | 10.8560 | -0.0592 | -0.2295 |
| | JJA | -0.0617 | 0.3630 | 13.7706 | 12.6158 | -0.3419 | -0.5571 |
| | SON | 0.1872 | 0.7033 | 22.0400 | 10.7076 | 0.0923 | -0.1999 |
| Munich | Annual | 0.0310 | 0.5964 | 43.9461 | 11.2703 | 0.3812 | -0.0682 |
| | DJF | -0.0408 | 0.5701 | 60.0883 | 15.9116 | 0.5412 | -0.0147 |
| | MAM | 0.0338 | 0.6998 | 53.9103 | 9.0543 | 0.5845 | -0.0188 |
| | JJA | -0.0714 | 0.3725 | 14.4982 | 9.0200 | -0.1439 | -0.3852 |
| | SON | 0.0447 | 0.5966 | 31.3270 | 9.7045 | 0.4013 | -0.0724 |
| Budapest | Annual | 0.2518 | 0.5404 | 22.9692 | 19.7840 | -0.2067 | -0.3728 |
| | DJF | 0.0837 | 0.5737 | 28.0308 | 22.3826 | -0.1267 | -0.3159 |
| | MAM | 0.1778 | 0.4994 | 21.4223 | 14.2831 | 0.0082 | -0.1924 |
| | JJA | 0.0900 | 0.2540 | 19.8013 | 20.5941 | -0.4760 | -0.6156 |
| | SON | 0.3210 | 0.6106 | 22.0720 | 20.9144 | -0.2610 | -0.3965 |
| Warsaw | Annual | 0.0882 | 0.6701 | 36.7845 | 16.7746 | 0.0725 | -0.2971 |
| | DJF | -0.0847 | 0.6332 | 53.4165 | 19.8648 | 0.2436 | -0.1765 |
| | MAM | 0.1073 | 0.6210 | 33.1155 | 15.3840 | 0.0906 | -0.2523 |
| | JJA | -0.0759 | 0.4087 | 19.8315 | 15.5113 | -0.2709 | -0.5781 |
| | SON | 0.0182 | 0.6840 | 32.6067 | 15.8365 | 0.1072 | -0.2739 |