# Peer review of "Modelling the European wind-blown dust emissions and their impact on PM concentrations"

_Atmospheric Chemistry and Physics, 2022_

## Referee Comment (RC1)

**Review for the manuscript submitted to ACP: Modelling the European wind-blow dust emissions and their impact on PM concentrations**

**General comments**

Dear editor and authors,

The manuscript attempts to quantify the wind-blow dust (WDB) emissions over Europe and their effect on PM2.5 and PM10 concentration by conducting simulation using CAMx coupled offline to WRF. The experiments span for a decade and are conducted with and without the WDB emissions over Europe. Additional experiments are conducted considering either ISORROPIA or EQSAM for secondary inorganic aerosol. The PM2.5 and PM10 are evaluated using stations from the AIRBASE network.

Overall, the manuscript is well written, the results in most cases are presented clearly and the topic is very relevant to ACP scope. However, based on the evaluation with AIRBASE, the addition WDB emissions over Europe and the produced PM deteriorates the performance of the model. In all cases the correlation drops with an increase of RMSE. Bias is rarely getting better, mainly over stations/seasons where PM were strongly underestimated in the noWDB experiments. As discussed in conclusions previous studies showed that the WDB emissions over Europe were by an order lower (Korcz et al., 2008) and the produced PM half (Vautard et al., 2005) of what is estimated by the current study. Considering the above I think it would be very beneficial to provide an uncertainty estimate for your WDB emissions over Europe that take into account (i) the overestimated wind and (ii) the combination of low LAI over urban grid boxes that can potentially be a false dust source. Further I have included some specific comments that can improve quality and readability of the manuscript as well as some technical corrections.

**Specific Comments**

- L126-127: Since there is a new reanalysis dataset (ERA-5) from the same source, do you think you would get different results if you used it to drive your model? Especially when considering that wind was overestimated in your simulations and that caused problems with the WDB emissions over Europe.
- Figure 2: "gs$^{-1}$" probably corresponds to "gram * second$^{-1}$" right? Or is it something else? Also it would be better to show these emission fluxes in km$^{-1}$ instead of gridbox$^{-1}$.
- L229-234: It is very valuable that you explain this clearly, though it unveils a potential bug for emissions over urban centers in Europe. To sum up your point, urban grid boxes at 9 x 9 km resolution in most cases are partially (<50%) characterized as only urban, while the rest is considered crop land with quite low leaf area index and thus potential dust sources. Would it be possible to fix that by setting the urban cover to 100% for grid boxes that are clearly cities or weight the LAI depending on the percentage of grid box characterized as crop land?
- L261 and Figure 7: Not in all stations see "Kralupy nad Vitavou"? The annual cycle of the measurements is very different in that case. Do we know why?

- L287-288: This could be easily checked by concentrating on these high peak days and evaluating the surface wind (average and max) with stations as well as checking if the wind-blown dust emissions are high?
- L305-308: Is this description better fitted in the Methods? Also note that RMSE is extra sensitive in outliers since the differences between the simulated and the observed values are squared.
- L319-321: Which means that the WBD emission scheme over Europe makes the model weaker in terms of PM2.5 and PM10, especially for correlations which in some cases it drops from 0.6 to 0.1 when WBD are considered. What is the main cause for that? I think you should propose potential causes for these results and discuss what can be improved in the current European WBD emissions set up in the model.
- L348: It would be a good addition at this point to explain why is that the case.
- L395-396: Since the WDB emissions over Europe is even smaller that you have estimated, there effect on PM is even smaller. Wouldn't that mean that your estimates are close to what has been reported before by Korcz et al. (2008) and Vautard et al. (2005)?

**Technical Corrections**

L19: "crustal" to "crystal"? If this right check it through out the text, e.g. L193, L197 etc.

L194: "surface temperature;" to "surface temperature,"?

L202: Could you rephrase please? "(only anthropogenic aerosol source and anthropogenic- and MEGAN-based gas-phase emissions)". E.g. "(including anthropogenic aerosol emissions as well as anthropogenic and biogenic gas-phase emissions)

---

## Author Comment (AC1)

**Authors response on the Anonymous Referee #3 review of "Modelling the European wind-blown dust emissions and their impact on PM concentrations" by**

Marina Liaskoni (acp-2022-804)

We thank Anonymous Referee #3 for his very detailed review and all the comments. We will address each of them and our point-by-point responses follow below. Reviewer's comments are *italicized*.

*General comment:*

*Overall, the manuscript is well written, the results in most cases are presented clearly and the topic is very relevant to ACP scope. However, based on the evaluation with AIRBASE, the addition WDB emissions over Europe and the produced PM deteriorates the performance of the model. In all cases the correlation drops with an increase of RMSE. Bias is rarely getting better, mainly over stations/seasons where PM were strongly underestimated in the noWDB experiments. As discussed in conclusions previous studies showed that the WDB emissions over Europe were by an order lower (Korcz et al., 2008) and the produced PM half (Vautard et al., 2005) of what is estimated by the current study. Considering the above I think it would be very beneficial to provide an uncertainty estimate for your WDB emissions over Europe that take into account (i) the overestimated wind and (ii) the combination of low LAI over urban grid boxes that can potentially be a false dust source. Further I have included some specific comments that can improve quality and readability of the manuscript as well as some technical corrections.*

Authors' response:
We agree with the reviewer, that due to the clear deterioration of model results when local wind-blown dust emissions are added, we have to analyse the sensitivities of the calculated emissions fluxes to the external data. We added, according to the suggestions of the reviewer, such analysis. Namely, we i) calculated the emissions fluxes for two wind reductions, 0.75 and 0.5 of the original wind values (given the seen overestimation of wind) and compared the obtained emission fluxes with the original ones as spatial plots seasonal emissions, i.e. new figures were added to the manuscript showing the new emissions fluxes after these modifications of input meteorological fields.

We also modified the way the LAI (or the vegetation factor, which is obtained from LAI) is averaged from the high resolution MODIS data for gridboxes with urban landuse fraction, see our answer further below.

*Specific comments*

*L126-127: Since there is a new reanalysis dataset (ERA-5) from the same source,*

*do you think you would get different results if you used it to drive your model? Especially when considering that wind was overestimated in your simulations and that caused problems with the WDB emissions over Europe.*

Authors' response:

Indeed, ERA-5 is a newer dataset and many comparison studies showed a better representation of measured wind-speeds in ERA-5 with respect to ERA-Interim. E.g. Belmore Rivas and Stoffelen (2019) showed higher near-surface wind-speeds in ERA-Interim over European seas which could imply higher winds than ERA-5 also on the boundaries of our domain which is over the Northern, Baltic and Mediterranean Seas Reduced wind-speeds then would imply reductions in WBD emissions. In the revised manuscript, this is briefly discussed (in the Discussion section).

*Figure 2: "gs-1" probably corresponds to "gram * second-1" right? Or is it something else? Also it would be better to show these emission fluxes in km-1 instead of gridbox-1 .*

Authors' response: yes, the units are grams per second (to avoid confusion, this is written out explicitly in the revised manuscript at the first appearance). We also changed the units from "per gridbox" to $km^{-2}$ all over the manuscript, including the plots.

*L229-234: It is very valuable that you explain this clearly, though it unveils a potential bug for emissions over urban centers in Europe. To sum up your point, urban grid boxes at 9 x 9 km resolution in most cases are partially (<50%) characterized as only urban, while the rest is considered crop land with quite low leaf area index and thus potential dust sources. Would it be possible to fix that by setting the urban cover to 100% for grid boxes that are clearly cities or weight the LAI depending on the percentage of grid box characterized as crop land?*

Authors' response: We agree with the reviewer that these low LAI values in the high resolution MODIS data within those model gridboxes where there is a significant urban landuse fraction can cause that the averaged LAI for the entire gridbox remain low causing emissions that otherwise would not appear (by getting below the threshold LAI value, 0.35). In our sensitivity test, we changed the way the LAI data are averaged within a model gridcell: from the averaging process (about 400 MODIS data points lie in a CAMx gridbox) we exclude the N lowest LAI values where N/400 is the urban fraction within the gridcell. In other words, we assume that the lowest (usually zero) LAI values are due to urban fraction. This process will not change the emissions for gridcells, where LAI is low due to other reasons, for example bare soil. Using the newly calculated emissions we plotted the spatial distribution in a similar fashion than Fig. 2 in the original manuscript and a clear reduction of emission fluxes near/around cities is obtained.

*L261 and Figure 7: Not in all stations see "Kralupy nad Vitavou"? The annual cycle of the measurements is very different in that case. Do we know why?*

Authors' response: The maximum values of the monthly average daily maxima are between January to March. For Kralupy nad Vltavou, it is clearly March and the same behavior is exhibited by stations in Brno and in Troubsko station while minima occur during late autumn to December in all stations. So in this regard, the annual cycle of the Kralupy nad Vltavou does not differ from other stations.

*L287-288: This could be easily checked by concentrating on these high peak days and evaluating the surface wind (average and max) with stations as well as checking if the wind-blown dust emissions are high?*

Authors' response: Yes, we agree that to check whether the strong overestimation of PM in model in connection with the peak concentrations relates to the wind bias, we have to check the modelled wind speed for the days when these peaks occur. Therefore, in the revised manuscript we included a new figure (Fig. 16 in the revised text) with the modelled/observed average monthly values of daily wind maxima but taken only for the days when the peaks occurred. This figure clearly shows that the wind-speed, both observed and modelled, are larger by about 50% for such days compared to the wind-speeds averaged for all days. The bias itself however remained the same in relative numbers, so the high positive wind bias is a systematic behavior throughout the year.

We also included another new figure - a scatter plot of the daily averaged WBD emissions from around Prague and PM concentrations modelled for the location of Prague stations to see how dust emissions are linked to high concentrations. We distinguished between concentrations below/above 100 ug/m3 to see if higher concentrations are more dependent on emissions. This figure confirms this and shows that for high emissions, there is a near linear relation to concentrations - in other words, high concentrations are due to WBD emissions.

*L305-308: Is this description better fitted in the Methods? Also note that RMSE is extra sensitive in outliers since the differences between the simulated and the observed values are squared.*

Authors' response: Although this description, agreeing with the reviewer, is usually in the Methods section, we prefer this to be placed here as these statistics are calculated only for PM. Hence we would like to place it in the PM validation subsection. Further we agree that RMSE strongly increases due to outliers and this has to be remembered when interpreting the results of the validation. Therefor we included a note on this in the revised manuscript.

*L319-321: Which means that the WBD emission scheme over Europe makes the model weaker in terms of PM2.5 and PM10, especially for correlations which in some cases it drops from 0.6 to 0.1 when WBD are considered. What is the main cause for that? I think you should propose potential causes for these results and discuss what can be improved in the current European WBD emissions set up in the model.*

Authors' response: yes, we agree that the model performance in terms of PM decreases strongly when WBD emissions are added to the simulation. This can be explained by the strong peaks in the impact on PM values which are a result of strong emission peaks seen in the daily time series of FCRS and CCRS emissions. The modelled urban PM peaks are often much higher (often by a factor of 5 or even more) than measurements and thus strongly reduce the correlation with the observed values. Also the RMSE values increased which can be again explained by the many outliers in the modelled PM data. This is detailed also in the Discussion section.

We also included there what should be improved to obtain a more accurate estimate of WBD emissions: it is high quality meteorological driving data especially with respect to wind fields, and consistent landuse and LAI data, preferably with comparable horizontal resolution.

*L348: It would be a good addition at this point to explain why is that the case.*

Authors' response: This is detailed in the Discussion section over lines 460-465 (with respect to the original manuscript). The reason for stronger sulphate and nitrate formation in EQSAM is most probably due to the fact, that in EQSAM, the cloud pH is influenced by three cations (Mg++, Ca++ and K+) while in ISORROPIA, it is only Ca++. This also explains the stronger decrease of ammonium in EQSAM which is replaced by more types cations (3 in EQSAM instead of 1 in ISORROPIA ).

*L395-396: Since the WDB emissions over Europe is even smaller that you have estimated, there effect on PM is even smaller. Wouldn't that mean that your estimates are close to what has been reported before by Korcz et al. (2008) and Vautard et al. (2005)?*

Authors' response: Yes, indeed. Our result might be strongly overestimated so it means that the effect on PM is very probably closer to Korcz and Vautard. We noted this in the revised manuscript in the Discussion section.

*Technical corrections*

*L19: "crustal" to "crystal"? If this right check it through out the text, e.g.*
*L193, L197 etc. L194: "surface temperature;" to "surface temperature,"?*

*L202: Could you rephrase please? "(only anthropogenic aerosol source and anthropogenicand MEGAN-based gas-phase emissions)". E.g. "(including anthropogenic aerosol emissions as well as anthropogenic and biogenic gas-phase emissions)*

Authors' response:

We meant crustal (not crystal), which refers to the Earth's crust, i.e. material that the Earth crust composed of. The semicolon was changed to comma and the sentence was rephrased to "including anthropogenic aerosol emissions as well as anthropogenic and biogenic gas-phase emissions"

References:

Belmonte Rivas, M. and Stoffelen, A.: Characterizing ERA-Interim and ERA5 surface wind biases using ASCAT, Ocean Sci., 15, 831–852, https://doi.org/10.5194/os-15-831-2019, 2019.

---

## Author Comment (AC2)

**Authors response on the Anonymous Referee #2 review of "Modelling the European wind-blown dust emissions and their impact on PM concentrations" by**

Marina Liaskoni (acp-2022-804)

We thank Anonymous Referee #2 for his comments. We address each of them and our point-by-point responses follow below. Reviewer's comments are *italicised*.

*General comments:*

*In most of the cases the highest impact of wind-blown dust on PM2.5, PM10 and PM components is observed in Berlin, Munich, Prague and Warsaw. The reasons for that should be stressed in the discussion part. What are the similarities/differences for those cities?*

Authors' response: Our emissions fields generated by the WBDUST produce high emission fluxes near each city in general, so this is not only the case of the mentioned cities but is evident also over e.g. the Ruhr area in Germany or over the Benelux states with dense urbanization. The reason is detailed in the manuscript and is that the MODIS data provides zero (or near zero) LAI values for urban areas. Then if there is a significant fraction of urban landcover within a model gridbox, the average LAI over the entire gridbox will be below the threshold (0.35) so the model evaluates the non-urban fraction of the gridbox as "WBD emitting", even if the LAI over the non-urban fraction (which are usually crops/forests) has LAI over the threshold. In the revised manuscript, we present a sensitivity test (new Fig. 6 in the revised manuscript) where, for such partly urbanized gridboxes, we averaged the MODIS LAI data in a way that we excluded the lowest LAI values that correspond to the urban fraction. This led to an evident decrease of emissions near cities.

The mentioned cities, in this regard, exhibit this behavior due to the fact that they are surrounded by forests so, due to what is mentioned above, the WBDUST model produces local dust emission peaks over these areas even if there should be no emissions or at least much lower emissions.

*Figs. 4, 5, 9-12, 13, 14, 15, 16, 20, 21, 22 should have the same scale of Y-axis among analyzed parameters. This will facilitate direct comparison of the analyzed values. Now, e.g. in Fig. 16 the maximum concentrations on Y-axis vary between 75 and up to 200.*

Authors' response: Yes, we agree that some of the plots should have the same scale of the Y-axis. We however want to keep different scales for the FCRS (fine mode dust) and CCRS (coarse mode dust) as the masses emitted in each of the modes differ greatly (which is logical). Specifically (referring the original manuscript Figure numbers - in the revision, new figures appeared so the numbering changed):

- Fig 3 and 5: for CCRS the scales are the same, but for the diurnal cycle of FCRS we have chosen a smaller scale to make the cycle more clear.
- Fig. 4: we kept the different Y-axis due to the above mentioned fact (FCRS vs CCRS)
- Fig. 9-10 will have a common Y-scale, Fig 11-12 will have also a common Y-scale, however due to the fact that PM10 values are much higher, this will be different from Fig. 9-10.

- Fig. 13: We unified the vertical Y-axis scale for all subplots. The same for Fig 14 (however, these are PM10 so different scale than for 13 which is PM2.5)
- Fig. 15: we unified the Y-scale for all PM2.5 impacts
- Fig. 16: we unified the Y-scale for all PM10 impacts
- Fig. 20-22: we unified the Y-scales, ranging from -1 to 1 (in case of PSO4, as only positive impacts are encountered, we set the range to 0-1).

*Specific comments:*

*Line 4: replace „wind-blow dust" by „wind-blown dust"*

*Line 193: replace „wind-blow dust" by „wind-blown dust"*

*Line 292: replace „wind-blow dust" by „wind-blown dust"*

*Line 340: replace „wind-blow dust" by „wind-blown dust"*

*Line 364: replace „looked at" by „aimed at"*

*Line 408: replace „bair soil" by „bare soil"*

*Line 429: replace „wind-blow dust" by „wind-blown dust"*

*Line 462: replace „Mg+" by „Mg++"*

Authors' response: All specific comments/corrections were implemented in the revised manuscript.

*Lines 687-694: 3 references appeared between Figs. 1 and 2, they should be placed before Fig. 1*

Authors' response: Resolved.